# Laboratory Validation of a Compact Single-Scattering Albedo (SSA) Monitor

Julia Perim de Faria[1], Ulrich Bundke[1,*], Andrew Freedman[2], Timothy B. Onasch[2], and Andreas Petzold[1]

[1]Forschungszentrum Jülich GmbH, IEK-8, 52425 Jülich, Germany

[2]Aerodyne Research, Inc., Billerica, MA 01821-3976, USA

*Correspondence to*: Ulrich Bundke (u.bundke@fz-juelich.de)

**Abstract**. An evaluation of the performance and relative accuracy of a Cavity Attenuated Phase-Shift Single Scattering Albedo Monitor (CAPS $PM_{ssa}$, Aerodyne Res. Inc.) was conducted in an optical closure study with proven technologies: Cavity Attenuated Phase-Shift Particle Extinction Monitor (CAPS $PM_{ex}$, Aerodyne Res. Inc.); 3-wavelengh integrating nephelometer (TSI Model 3563); and 3-wavelength filter-based Particle Soot Absorption Photometer (PSAP, Radiance Research). The evaluation was conducted by connecting the instruments to a controlled aerosol generation system and comparing the measured scattering, extinction, and absorption coefficients measured by the CAPS $PM_{ssa}$ with the independent measurements. Three different particle types were used to generate aerosol samples with single-scattering albedos (SSA) ranging from 0.4 to 1.0 at 630 nm wavelength. The CAPS $PM_{ssa}$ measurements compared well with the proven technologies. Extinction measurement comparisons exhibited a slope of the linear regression line for the full data set between 1.05 and 1.01, an intercept below $\pm 1.5 \times 10^{-6}$ m$^{-1}$ ($\pm 1.5$ Mm$^{-1}$), and a regression coefficient $R^2 > 0.99$; whereas, scattering measurements had a slope between 0.90 and 1.04, an intercept of less than $\pm 2.0 \times 10^{-6}$ m$^{-1}$ (2.0 Mm$^{-1}$), and a coefficient $R^2 > 0.96$. The derived CAPS $PM_{ssa}$ absorption compared well to the PSAP measurements for the small particle sizes and modest (0.4 to 0.6) SSA values tested, with a linear regression slope between 0.90 and 1.07, an intercept of $\pm 3.0 \times 10^{-6}$ m$^{-1}$ (3.0 Mm$^{-1}$), and a coefficient $R^2 > 0.99$. For the SSA measurements, agreement was highest (regression slopes within 1%) for SSA = 1.0 particles at extinction levels of tens of Mm-1 and above; however, as extinctions approach zero, small uncertainties in the baseline can introduce larger errors. SSA measurements for absorbing particles exhibited absolute differences up to 18%, though it is not clear which measurement had the lowest relative accuracy. For a given particle type, the CAPS $PM_{ssa}$ instrument exhibited the lowest scatter around the average. This study demonstrates that the CAPS $PM_{ssa}$ is a robust and reliable instrument for the direct measurement of the scattering and extinction coefficients and thus SSA. This conclusion also holds as well for the indirect measurement of the absorption coefficient with the constraint that the relative accuracy of this particular measurement degrades as the SSA and particle size increases.

Keywords: CAPS $PM_{ssa}$, optical closure, single scattering albedo.

## 1 Introduction

Airborne aerosols impact climate directly though the interaction with incident solar light by scattering, generating a cooling effect, or by absorbing it and reemitting infrared radiation, having a heating effect. According to Haywood and Shine (1995), the effect of aerosols on the atmospheric radiation budget in the visible spectral range depends on the aerosol optical depth (AOD), the single-scattering albedo (SSA), and the backscattered fraction (BF). The radiative forcing efficiency (RFE) describes the resulting aerosol direct forcing per unit AOD (Haywood and Shine, 1995;Andrews et al., 2011;Sheridan et al., 2012) and is widely used to describing the radiative impact of a given aerosol type. As an aerosol

intensive parameter the RFE value depends only on SSA and BF. As is stated in the latest IPCC report (Boucher et al., 2013), uncertainties in SSA and the vertical distribution of aerosol contribute significantly to the overall uncertainties in the direct aerosol radiative forcing, while AOD and aerosol size distribution are relatively well constrained.

The measurement of SSA requires the simultaneous but independent observation of two parameters since, by definition, the SSA is the ratio of the scattering to the extinction coefficient (where extinction is the sum of the scattering and absorption – see Equation (1) and (2); the index p refers to the contribution of aerosol particles to overall light extinction, which has also a contribution by gas molecules, identified by the index g not shown in the equation).

$$\sigma_{ep} = \sigma_{ap} + \sigma_{sp} \tag{1}$$

$$SSA = \sigma_{sp}/\sigma_{ep} \tag{2}$$

Measuring all three aerosol optical coefficients independently allows for the closure of optical properties and thus the determination of the relative uncertainties of the involved instruments.

The aerosol optical parameters are typically measured *in-situ* by instruments such as integrating nephelometers (NEPH) for the scattering coefficient (Heintzenberg and Charlson, 1996); photoacoustic (see e.g., Lack et al. (2006); Arnott et al. (2006)) and filter-based methods such as the Particle-Soot Absorption Photometer (PSAP; Bond et al. (1999)), the

50 Multi Angle Absorption Photometer (MAAP; Petzold and Schönlinner (2004)) and more recently the Tricolor Absorption Photometer (TAP/CLAP; Ogren et al. (2017)) for the absorption coefficient; and for the extinction coefficient, the cavity ring down (CRD) technology (Moosmüller et al., 2005) or, since 2007, the Cavity Attenuated Phase Shift Particle Extinction Monitor (CAPS PM$_{ex}$) (Massoli et al., 2010). To measure the SSA using the optical closure approach involves separate instruments with different principles and uncertainties, leading to potential sources of significant errors and biases.

A novel instrument based on cavity attenuated phase-shift technology and incorporating an integrating sphere was recently developed by Aerodyne Research, Inc. This novel instrument represents a major step forward in the observation of aerosol optical properties since it simultaneously measures two of the three aerosol optical parameters from the same air sample, reducing the potential sources of sampling biases (Onasch et al., 2015). The two main applications of the CAPS PM$_{ssa}$ instrument, apart from the direct measurement of scattering and extinction coefficients, are the indirect measurement

of the aerosol absorption coefficient and the measurement of the single-scattering albedo. A few recent *in-situ* application studies of the CAPS PM$_{ssa}$ instrument are already available (Han et al., 2017;Corbin et al., 2018). The present optical closure study intends to quantify relative uncertainties in the measurement of the primary aerosol optical properties and the resulting SSA by the CAPS PM$_{ssa}$ for several types of laboratory aerosol by applying a full set of established instrumentation for measuring the extinction (CAPS PM$_{ex}$), absorption (PSAP), and scattering (integrating nephelometer

TSI Model 3563) coefficients at multiple wavelengths.

## 2 Instruments and Methods

### 2.1 Instrumental Set-up

The laboratory study was conceived to evaluate the operational principle of the CAPS PM$_{ssa}$ and its performance and relative accuracy when compared to proven technologies. The instrumental set-up used is shown in Figure 1.

In this study, similar to previous work (Massoli et al., 2010; Petzold et al., 2013); two collison-type aerosol generators (TSI Model 3076) were used; one containing a solution of deionized water and purely scattering aerosol, ammonium sulphate (AS), and a second containing absorbing aerosol, water-soluble colloidal graphite (Aquadag – AD – from Agar Scientific) or black carbon (REGAL 400R Pigment Black – BC – from Cabot Corporation). The SSA of the

dispersed aerosol ranged from approximately 0.4 (pure AD or BC) to 1.0 (pure AS), with the modal value of the particle size distribution being below 100 nm in all cases. A drying tube filled with silica gel was positioned after each particle generator in order to reduce the relative humidity below 30%. Once the samples were passed through the dryer, they entered a mixing chamber where effective ensemble particle SSA values of 0.4 < SSA < 1.0 could be produced by mixing aerosol flows containing both absorbing and scattering aerosols. The aerosol generation set-up specifications are shown in Table 1, whereas Table 2 compiles the information about the applied instruments and correction schemes. The SSA of the mixture containing AS and AD was controlled by the online measured SSA measured by the CAPS PM$_{ssa}$.

Five mass flow controllers (MFC), two at each generator's head and a third after the mixing chamber, supplied particle-free compressed air to the sample to both reach the desired humidity and particle number concentration and to make-up the flow required by the instruments. The particle number concentration was measured by a condensation particle counter (CPC).

**Table 1. Type of generated aerosol, targeted SSA (630 nm), and targeted max. aerosol extinction values**

| Aerosol type | Estimated /Expected SSA | Run 1 200 Mm$^{-1}$ | Run 2 150 Mm$^{-1}$ | Run 3 100 Mm$^{-1}$ | Run 4 50 Mm$^{-1}$ | Run 5 25 Mm$^{-1}$ |
|---|---|---|---|---|---|---|
| Aquadag (AD) | 0.4 | x | x | x | x | x |
| Black Carbon (BC) | 0.4 | | x | x | x | x |
| Mixture (AS+AD) | 0.6 | | | x | x | x |
| Ammonium Sulphate (AS) | 1.0 | | x | x | x | x |

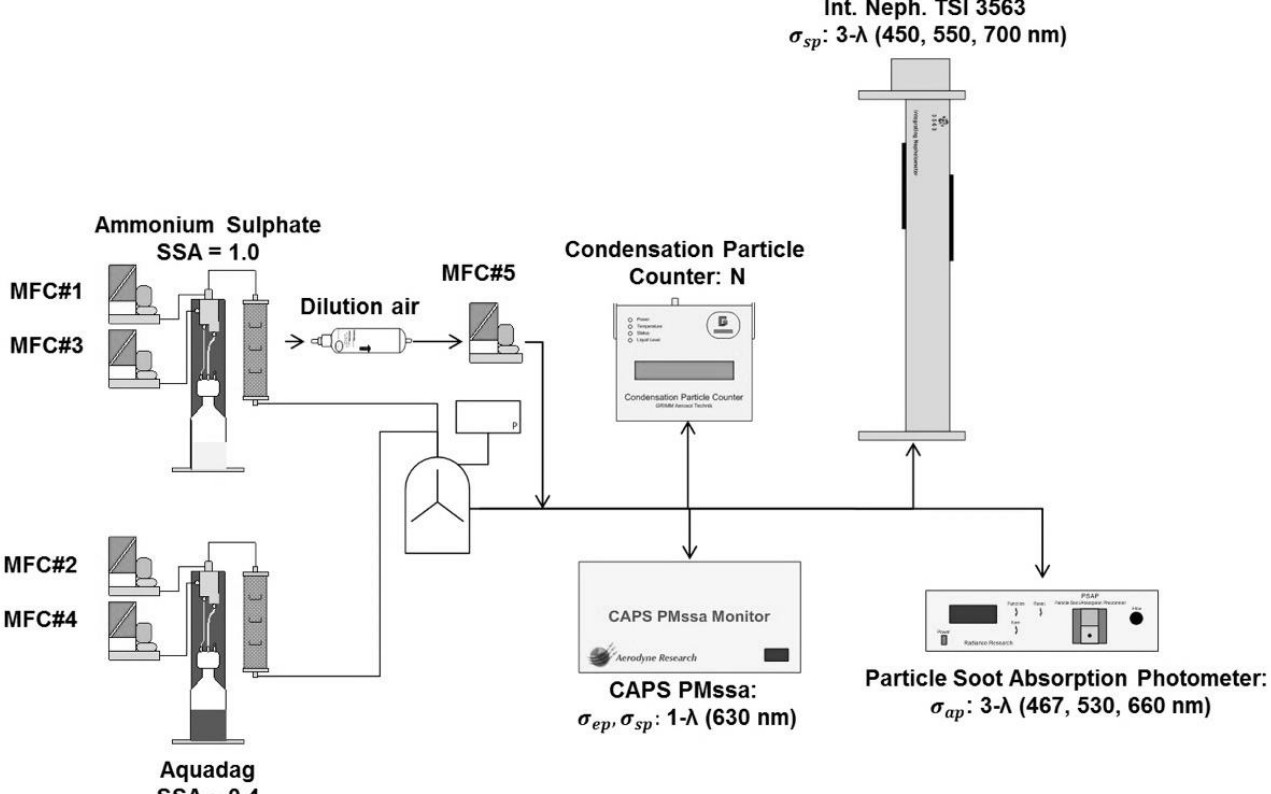

**Figure 1. Instrumental set-up applied in the optical closure study**

**Table 2. List and specifications of optical instrumentation and applied correction algorithms**

| Instrument | Manufacturer | Property | $\lambda$ (nm) | Aerosol | Correction Algorithm |
|---|---|---|---|---|---|
| CAPS PM$_{ssa}$ | Aerodyne Research Inc. | $\sigma_{sp}$, $\sigma_{ep}$ | 630 | AS, AD, BC, MIX | Mie Amigo (Aerodyne) for $\sigma_{sp}$ truncation correction (Onasch et al., 2015) |
| CAPS PM$_{ex}$ | Aerodyne Research Inc. | $\sigma_{ep}$ | 630 | AS, AD, BC, MIX | No correction required |
| NEPH | TSI Inc. | $\sigma_{sp}$ | 450, 550, 700 | AS | Müller et al. (2009), Anderson et al. (1998) |
| | | | | AD, BC, MIX | Massoli et al. (2009) |
| PSAP | Radiance Research Inc. | $\sigma_{ap}$ | 467, 530, 660 | AS, AD, BC, MIX | Ogren (2010) and Virkkula (2010) |

The samples were produced at up to five nominal concentration levels, as shown in Table 1, defined by the aerosol extinction. This was achieved by holding the aerosol generation system constant (MFC#1 - MFC#4) and regulating the diluting air MFC#5). Extinction coefficient levels were varied from ~10 up to 200 Mm$^{-1}$. For each level, a sampling time of at least 5 minutes was sustained.

To ensure an isoaxial, isokinetic sampling by all instruments, special sampling tips made of stainless steel were designed such that the sample air extraction tips were each concentrically placed along the centre line of the sample tube of 1 inch inner diameter. The inlet nozzles diameters are dimensioned such that the flow velocities in the sample tube and inside extraction tip nozzles match. Distances between the extraction points for the different instruments were 20 cm.

The nephelometer was calibrated using $CO_2$ (high span gas) and particle-free air (low span gas), before starting the experiments. The calibration procedure also includes, as recommended by the manufacturers, the calibration of the scattering channel of the CAPS PM$_{ssa}$ against the extinction channel of the instrument. For the filter-based absorption instruments, no calibration is necessary since they both operate with a blank filter in parallel as reference (see description in the subsections below).

The optical instruments were placed downstream from the generation system, measuring simultaneously, as shown, and will be described in more detail in the following subsections.

### 2.1.1    Integrating Nephelometer

In this optical closure study, an integrating nephelometer (NEPH) of the type TSI Model 3563 was used. The NEPH collects scattering measurements both in the forward and backscatter directions at three wavelengths 450, 550, and 700 nm (Heintzenberg et al., 2006).

The NEPH data were corrected for truncation angle effects using the approach proposed by Massoli et al. (2009) for strongly light-absorbing aerosols (equations 3 and 4 and Table 3). For predominantly light-scattering aerosols, the approaches proposed by Anderson et al. (1996) and Müller et al. (2009) were used (equation 5 and Table 4).

$$C = MAX\{1.0, v_0 + v_1 \exp(v_2 * (3.25 - \text{å})) + C(n)\} \tag{3}$$

where å is the Ångstrom exponent (equation 8), $C(n)$ is an optional correction for submicron distributions. $C(n)$ is equal to 0 for å ≥2.8, and to

$$C(n) = v_3(2.8 - \text{å}) * \left(\frac{1}{(n-1)} - \frac{1}{0.48}\right) \tag{4}$$

and $v_0, v_1, v_2$ and $v_3$ are given in Table 3 and n is the real part of the refractive indices.

**Table 3. Coefficient values for $v_0, v_1, v_2$ and $v_3$ for equations 3 and 4 (Massoli et al., 2009).**

|  | $v_0$ | $v_1$ | $v_2$ | $v_3$ |
|---|---|---|---|---|
| 698 nm sub-µm | 0.8627 | 0.1423 | 0.1816 | 0.0306 |
| 554 nm sub-µm | 0.8511 | 0.1589 | 0.2153 | 0.0439 |
| 453 nm sub-µm | 0.8863 | 0.1327 | 0.2758 | 0.0610 |
| 698 nm all | 0.9869 | 0.0182 | 0.7980 | |
| 554 nm all | 0.9948 | 0.0152 | 0.8951 | |
| 453 nm all | 1.0072 | 0.0118 | 1.0036 | |

$$C = a + b * \text{å} \tag{5}$$

 **Table 4. Values for a and b for equation 5 for Anderson and Ogren (1998) and Müller et al. (2011).**

|  |  | Blue (450 nm) | | Green (550 nm) | | Red (700 nm) | |
|---|---|---|---|---|---|---|---|
| **Ångstrom exponent** |  | å(B/G) | | å (B/R) | | å (G/R) | |
|  |  | a | b | a | b | a | b |
| Anderson et al. (1998) | No cut | 1.365 | -0.156 | 1.337 | -0.138 | 1.297 | -0.113 |
|  | Sub-µm | 1.165 | -0.046 | 1.152 | -0.044 | 1.120 | -0.035 |
| Müller et al. (2011) | No cut | 1.345 | -0.146 | 1.319 | -0.129 | 1.279 | -0.105 |
|  | Sub-µm | 1.148 | -0.041 | 1-137 | -0.040 | 1.109 | -0.033 |

### 2.1.2 Particle-Soot Absorption Photometer

The PSAP is a filter-based three wavelength (467, 530, 660 nm) instrument, manufactured by Radiance Research, that provides continuous measurement of the light absorption coefficient. The instrument uses two spots on a quartz fibre filter; one receives the particle containing sample and the second clean air. The instrument measures the difference in the transmission of light between a loaded and a blank filter spot (Bond et al., 1999). Two absorption coefficient data corrections were used and evaluated: Ogren (2010) and Virkkula (2010). The best fitting correction is the one shown in each result subsection.

In his approach, Ogren (2010) furthers the corrections from Bond et al. (1999), considering the filter area correction and wavelength adjustment, as shown in equation 6, for the complete absorption coefficient measurement.

$$\sigma_{ap} = 0.85 \left(\frac{Q_{PSAP}}{Q_{meas}}\right)\left(\frac{A_{meas}}{A_{PSAP}}\right)\frac{\sigma_{PSAP}[\lambda]}{K_2} - \frac{K_1}{K_2}\sigma_{sp}[\lambda] \tag{6}$$

where $\sigma_{ap}$ is the absorption coefficient of the desired wavelength, $Q_{PSAP}$ is the flow recorded by the instrument, $Q_{meas}$ is the measured flow, $A_{meas}$ is the real area of the filter, $A_{PSAP}$ is the manufacturer supplied area of the filter, $\sigma_{PSAP}$ is the measured absorption coefficient at a certain wavelength ($\lambda$), $K_1$ and $K_2$ are constants given (0.02 ± 0.02 and 1.22 ± 0.20, respectively) and $\sigma_{sp}$ is the scattering coefficient measured at the same wavelength as $\sigma_{PSAP}$.

Virkkula (2010) derives a new correction from a field campaign, including as a function factor the single scattering albedo, as shown in equation 7.

$$\sigma_{ap} = (k_0 + k_1(h_0 + h_1\omega_0)\ln(Tr))\sigma_{PSAP}[\lambda] - s\sigma_{sp}[\lambda] \qquad (7)$$

where $\sigma_{ap}$ is the absorption coefficient of the desired wavelength, $k_0$, $k_1$, $h_0$, $h_1$ and $s$ are constants given (Table 5), $\omega_0$ is the single scattering albedo, Tr is the measured transmission, $\sigma_{PSAP}$ is the value measured by the PSAP and $\sigma_{sp}$ is the scattering coefficient measured at the same wavelength as $\sigma_{PSAP.}$

**Table 5. Constant values given by Virkkula (2010) for equation 7**

| Constant | 467 nm | 530 nm | 660 nm |
|---|---|---|---|
| $k_0$ | 0.377 ± 0.013 | 0.358 ± 0.011 | 0.352 ± 0.013 |
| $k_1$ | -0.640 ± 0.007 | -0.640 ± 0.007 | -0.674 ± 0.006 |
| $h_0$ | 1.16 ± 0.05 | 1.17 ± 0.03 | 1.14 ± 0.11 |
| $h_1$ | -0.63 ± 0.09 | -0.71 ± 0.05 | -0.72 ± 0.16 |
| $s$ | 0.015(0.009, 0.020) | 0.017(0.012, 0.023) | 0.022(0.016,0.028) |

### 2.1.3    The CAPS PM$_{ex}$

The CAPS PM$_{ex}$ system, described in detail and assessed in several studies, such as Massoli et al. (2010), Petzold et al. (2013) and Perim de Faria et al. (2017) measures light extinction by determining the change in signal phase shift caused by the introduction of particles into an optical cavity. The use of high reflectivity mirrors (reflectivity approximately 99.99%) in the optical cavity creates the long measurement path of approximately 2 km required to measure very low values of light extinction (limit of detection of 1-2 Mm$^{-1}$ in 1 second sample period).

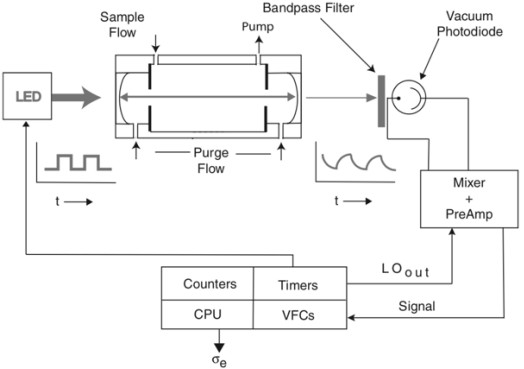

**Figure 2. Overview of the main components and operation principle of the CAPS PM$_{ex}$ instrument (Massoli et al., 2010)**

### 2.1.4    The CAPS PM$_{ssa}$

The CAPS PM$_{ssa}$ (Onasch et al., 2015), uses the same principle to measure light extinction as the CAPS PM$_{ex}$, but it also contains, located at the centre of the measurement cell, a 10 cm diameter integrating sphere capable of measuring light scattering on the same aerosol sample, as shown in Figure 3. The integrating sphere acts as an integrating nephelometer, which measures the scattering of light by particles at all angles, only excluding the near 0 and near 180° angles since the opening of the extinction chamber is located in these directions, allowing the sample and light beam to pass through. The sphere shows 98-99% Lambertian reflectance efficiency due to its high reflectivity coating (Avian D from Avian Technologies). The usage of an integrating sphere increases the collection of scattered light at the photomultiplier compared to a traditional cosine corrected detector arrangement.

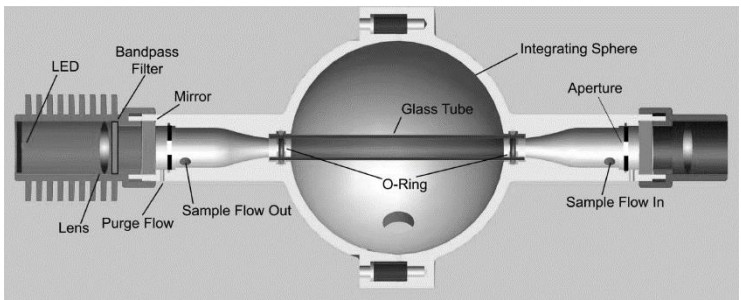

**Figure 3. CAPS PMssa components and set-up (Onasch et al., 2015).**

The scattering channel is calibrated against the extinction channel using either small particles (<250 nm) that have SSA=1.0 or $CO_2$ and set equal to the extinction measurement. This calibration procedure also allows the user to prove monitor linearity over a wide range of optical extinctions without the limitation of using individual gases normally used for nephelometer calibrations. Checking the calibration by using $CO_2$ we observed neglectable differences.

The sample flow in the instrument is set to 0.85 lpm and is controlled by a critical orifice. The measurement sample enters the chamber in one end and exits through an opening located in the other end flowing through a glass tube inside the integrating sphere (Figure 3). The mirrors are kept particle-free by a continuously flowing purge flow (25 $cm^3$ $min^{-1}$). Petzold at al. (2013) showed that this purge flow shortens the measurement path and dilutes the sample and requires a correction factor. As done for the CAPS PM$_{ex}$, a new correction factor was developed, by using monodisperse polystyrene latex spheres (PSL) of known size, for the CAPS PM$_{ssa}$. Due to the cell geometry, the new correction factor was slightly larger than the one found for the extinction monitor, 1.37 and 1.27, respectively (Onasch et al., 2015). The noise of the instrument, trunction angle, and instrument uncertainty have also been studied by Onasch et al. (2015). The values found were all below 1 $Mm^{-1}$ for the noise levels (1σ, 1s) for all wavelengths. For the case of this particular instrument (630 nm), the truncation correction was determined to be lower than 4% for typical ambient conditions. The uncertainty was estimated at ±0.03 for SSA equal to 1 (PSL and ammonium sulphate) and decreases to ±0.01 as the SSA goes down.

The baseline determination system is identical to the one used in the CAPS PM$_{ex}$, in which filtered and thus particle-free sample air fills the measurement chamber and is used to quantify contributions of gas molecules to the instrument response by Rayleigh scattering and potential absorption of light, and to determine interferences of system components. Both the CAPS PM$_{ex}$ and CAPS PM$_{ssa}$ used in this study operate at a wavelength of 630 nm and thus show minimal interference from absorption by ambient gaseous species like $NO_2$ and $H_2O$.

## 2.2 Data Treatment

All multi-wavelength instruments were adjusted to match the other instruments' wavelengths for the intercomparison by using the Ångström exponent approach; see Equation (8) and (9),

$$å = - \frac{log \frac{\sigma_x}{\sigma_y}}{log \frac{x}{y}} \qquad (8)$$

$$\sigma_w = \sigma_y \times \left(\frac{w}{y}\right)^{-å} \qquad (9)$$

where å is the Ångström exponent, σ is the optical property measured (extinction, scattering or absorption coefficient), x and y are the operating wavelengths of the instrument, and w refers to the wavelength, to which the property should be adjusted. For a better understanding of the wavelength adjustment, the complete description is given in Figure 3 from Petzold et al. (2013).

All instruments provide 1 second resolution data. Data were collected over 5 minutes for each experimental point to remove any effect of differences in response times and fluctuations in the aerosol generation system. In Figure 5 the data were averaged for each extinction/scattering/absorption level, and the standard deviation was calculated from the mean.

Standard linear regression analysis was performed for the 10 seconds average of the data set.

### 2.3 Measurement uncertainties

This paper does not address in any explicit way, nor was it designed to address, the question of the absolute uncertainties of the different measurement techniques. It was designed to address the question of how well they correlate. Thus, the results are given in correlation coefficients (slope and intercept) and their statistical uncertainties.

For this reason, this section compiles the reported relative uncertainties of the accuracy by the relevant instrument papers.

**Table 6. Measurement relative uncertainties of the accuracy for the different instruments as reported by the relevant instrument papers.**

| Instrument | SEP | SSP | SAP | SSA | Reference | Comments |
|---|---|---|---|---|---|---|
| CAPS PM$_{SSA}$ | 5% | 8% | 13% (SSA=0.5) 5%(SSA=1.0) | 3% | (Onasch et al., 2015) | Estimates for polydisperse aerosol. Absorption uncertainty is dependent upon the SSA value |
| NEPH | | <10% | | | (Anderson et al., 1996); (Massoli et al., 2010) | For submicrometer particles |
| PSAP | | 8% | | | (Muller et al., 2014) | |
| NEPH+PSAP | 7% | | | | (Petzold et al., 2013) | 3-sigma obtained for the test aerosol inversion of NEPH+PSAP data |

Using the reported relative uncertainties of the accuracy in Table 6, we calculated the derived uncertainties using Gaussian error propagation for SSA and the absorption coefficient for all instrument combinations. The formulas are derived in Appendix A and the associated graphs Figure 13 until Figure 15 are shown. They are summarized as follows: for the CAPS PM$_{SSA}$ Monitor the relative error (as defined by (rel_err(x)=Δx/x) for the oberservable x={{$\sigma_e$,$SSA$}, $\sigma_e$}) depends on the SSA and on the aerosol load of the test aerosol as stated by (Onasch et al., 2015). This dependency is best visible during transitions of the aerosol production system, where the SSA varies with time and where the particle load is low associated with scattering coefficients of about 10 Mm$^{-1}$. Here the relative error ranges within 6-13% for 1Hz data (2-4% for 10 second averaged data). For high aerosol loads the relative error ranges within 8%-10% for 1 Hz data( 2.5%-3% for 10 second averaged data) For the absorption coefficient derived from the CAPS SSA monitor the relative uncertainty rises with higher SSA values from 8% (SSA=0.25) up to 25% (SSA= 0.65) for 1 Hz data (2.5% -8% for 10 second averaged data respectively).

The relative errors analysis of the SSA shows that the CAPS PM$_{ssa}$ instrument is less sensitive to the aerosol load (8% 1Hz data and 2.5% 10 second averaged data) compared to the proven PSAP+NEPH instrument combination which shows a pronounced dependence but is principle more robust towards low aerosol load.

## 3  Results and Discussion

In this section, we present the results and relevant discussion of findings for the optical closure study. All the measurements presented here were corrected to the CAPS PM$_{ssa}$ operational wavelength of 630 nm.

### 3.1  Extinction Coefficient

The extinction coefficient measured by the CAPS PM$_{ssa}$ was analysed in comparison with proven technologies. On the direct measurement of $\sigma_{ep}$, we compared the two CAPS systems for AS and AD (Petzold et al., 2013). The direct measurement of $\sigma_{ep}$ from the CAPS PM$_{ssa}$ was also compared with the indirect measurement given by the sum of the absorption coefficient measured by the PSAP with the scattering coefficient measured by the NEPH for BC, AD, and MIX (as defined in Table 1) – shown as PSAP+NEPH. For AS with the measured SSA value of 1.0, extinction coefficients provided by the CAPS extinction channels and scattering coefficients provided by the CAPS scattering channel and the NEPH instrument are used for the evaluation of the light scattering measurements in the next subsection. The time series for the extinction channels are shown in Figure 4 and the averages and standard deviations for each test point are shown in Table A1 in the supplemental information. The higher variability observed in the last plot of the figure is due to particle load fluctuations from the generation system when operating at very high loads.

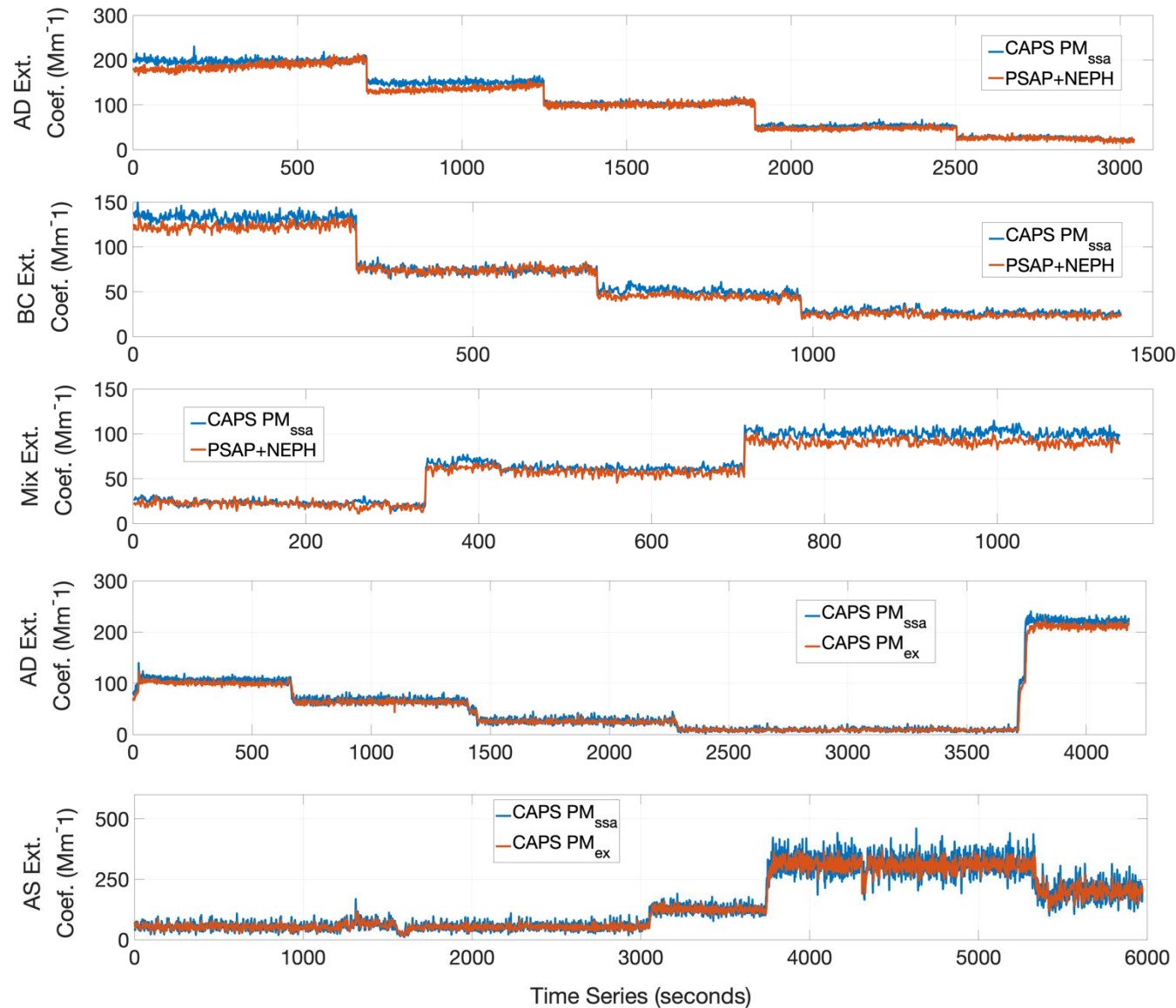

**Figure 4. Time series of the parallel measured extinction coefficients by the different instruments. Instruments used: CAPS PM$_{ex}$, CAPS PM$_{SSA}$ and the sum of absorption- and scattering coefficients measured by PSAP and NEPH as noted in the legend for the individual subplots. The test aerosols used are noted in the caption of the y-axis of the individual subplots.**

Figure 5 shows the scatter plot of the average value and standard deviation for each level of measured extinction coefficient for the two CAPS systems for AD and AS and the comparison with the sum of the NEPH and PSAP for AD and BC. The best results for the AD and BC were found when applying the Massoli et al. (2009) correction with the assumption, that no particle size cut has been used for the inlet system (no-cut approach) to the NEPH data, and Virkkula (2010) for strongly light-absorbing aerosols AD and BC to the PSAP data. For the mixture, the applied corrections were Anderson et al. (1998) for the NEPH data and Ogren (2010) for the PSAP data. The extinction channels from the two CAPS and the sum of the NEPH and PSAP (PSAP+NEPH) signals show a good agreement for all aerosol types, with linear regression (10s average data) slopes (m) between 1.01 and 1.06, offsets (b) bellow 1.1 Mm$^{-1}$ and correlation coefficients above 0.99. The slopes of the regression analysis of the 10 second averaged data (see Figure 6 as an example) and their standard deviation are shown in Table 7 as a function of the sampled aerosol type and associated single-scattering (SSA) albedo. As it can be seen there is no systematic difference in the slope with increase or decrease of the aerosol SSA.

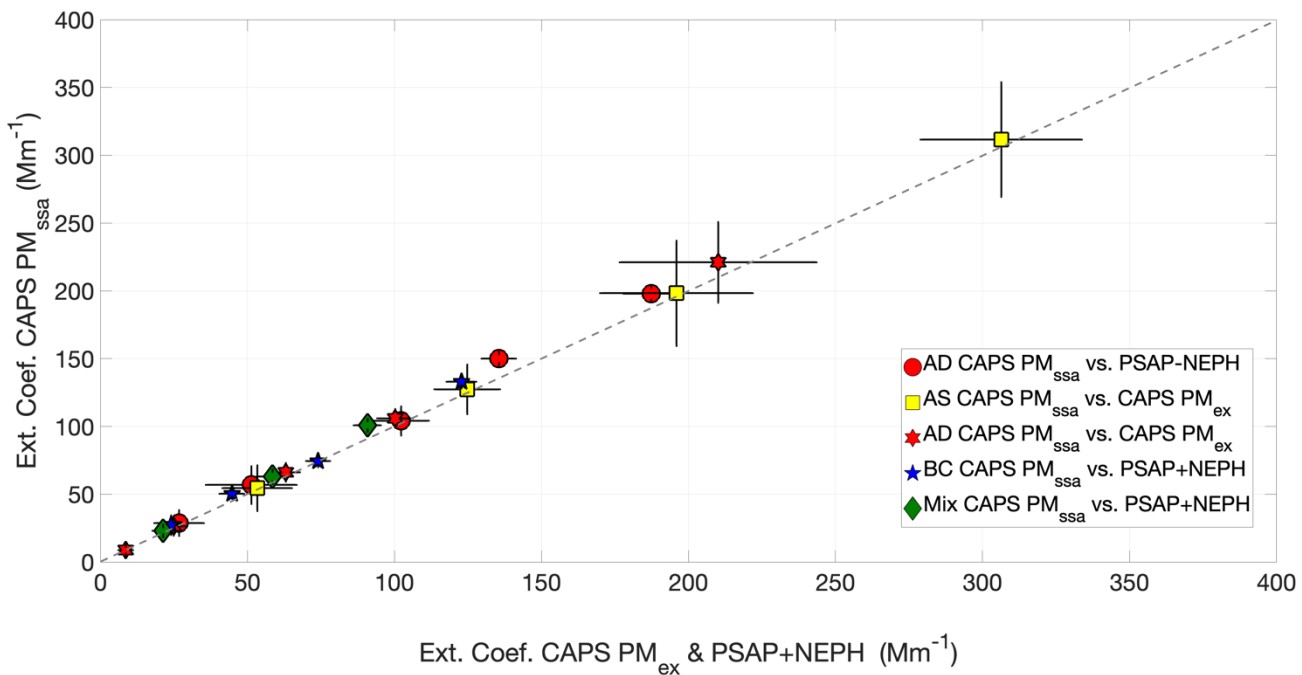

**Figure 5. Comparison result (mean value and standard deviation for each level) of the extinction channel of the CAPS PM$_{ssa}$ with the CAPS PM$_{ex}$ and the combination PSAP+NEPH for the different aerosol types (dashed line is the identity line (i.e., 1:1 line)).**

**Table 7 Linear regression parameters obtained by fitting 10 second averaged data including the slope (M), standard deviation of the slope (Std m), intercept (B), standard deviation of intercept (std b), and linear regression coefficient (R$^2$) for the comparison of the CAPS PM$_{ssa}$ extinction channel with proven technologies**

| Aerosol | Reference Instrument | Nominal SSA | M | Std m | B | Std b | R$^2$ |
|---------|---------------------|-------------|------|-------|-------|-------|------|
| AD | PSAP+NEPH | 0.4 | 1.05 | 0.00 | 0.03 | 0.08 | 0.99 |
| BC | PSAP+NEPH | 0.4 | 1.06 | 0.00 | 1.07 | 0.17 | 0.99 |
| MIX | PSAP+NEPH | 0.6 | 0.99 | 0.00 | -0.72 | 0.15 | 1.00 |
| AD | CAPS PM$_{ex}$ | 0.4 | 1.05 | 0.00 | 0.03 | 0.08 | 1.00 |
| AS | CAPS PM$_{ex}$ | 1.0 | 1.01 | 0.00 | 1.02 | 0.26 | 0.99 |
| All | CAPS PM$_{ex}$ | NA | 1.01 | 0.00 | 1.24 | 0.15 | 0.99 |

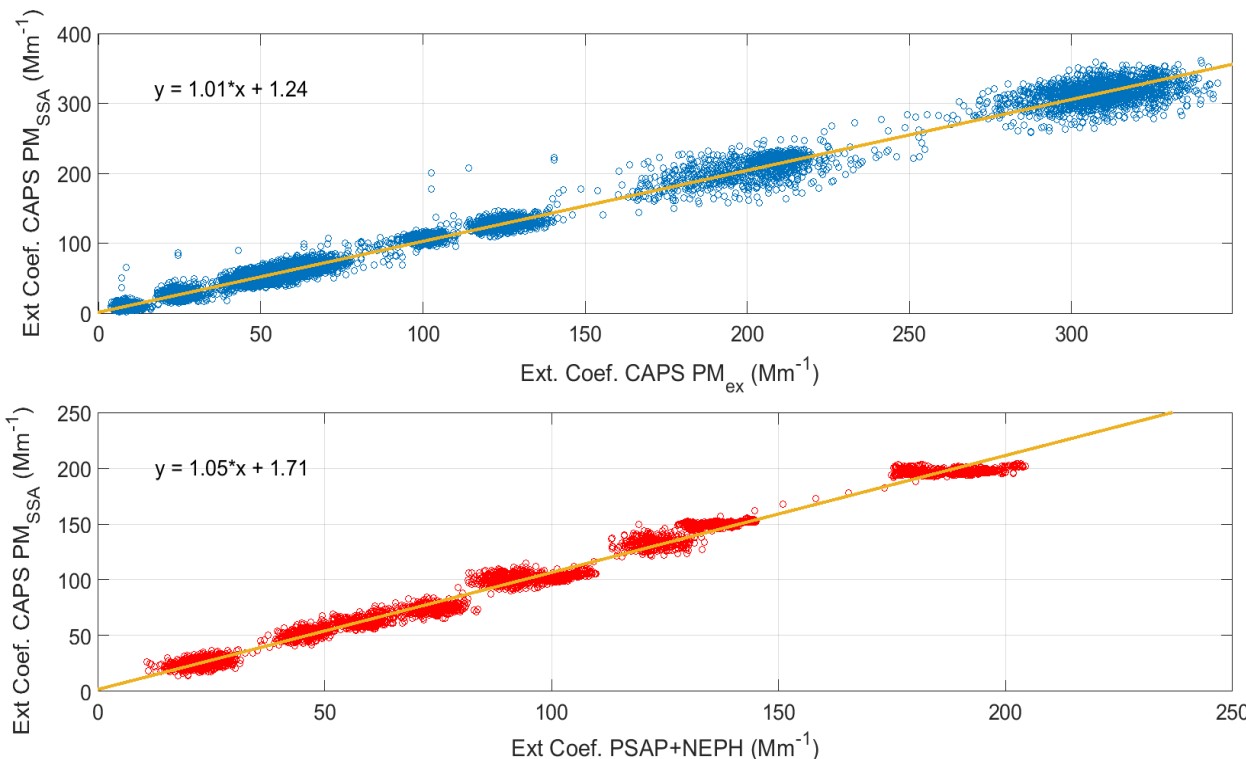

**Figure 6. Scatterplot and regression line and equation of the extinction coefficient measured by the CAPS PM$_{SSA}$ and by proven technologies for all tested aerosols for the 10s average data.**

It is worth noting that for the particular instruments used in our study, the standard deviation for the extinction data of the CAPS PM$_{ssa}$ is larger than for the extinction data provided by the CAPS PM$_{ex}$ (horizontal error bars). This finding is shown in the histogram of the extinction channel from one measurement level (in this case the used dataset refers to the 25 Mm$^{-1}$ target-level for AD aerosol) for both equipment (Figure 7). Thus, the precision of this particular CAPS

PM$_{ssa}$ is lower than the precision of the CAPS PM$_{ex}$. Regarding the precision of the CAPS PM$_{ssa}$ in comparison with proven technologies, the standard deviation found in this study for both cases are comparable. The precision in the CAPS PM$_{ex}$ and PSAP+NEPH extinction measurements found in this study are very similar to the one found by Petzold et al. (2013), in which an excellent correlation (slope of 0.99) was found for the laboratory comparison between the same instruments using highly absorbing aerosol, exclusive scattering aerosol and mixtures of both.

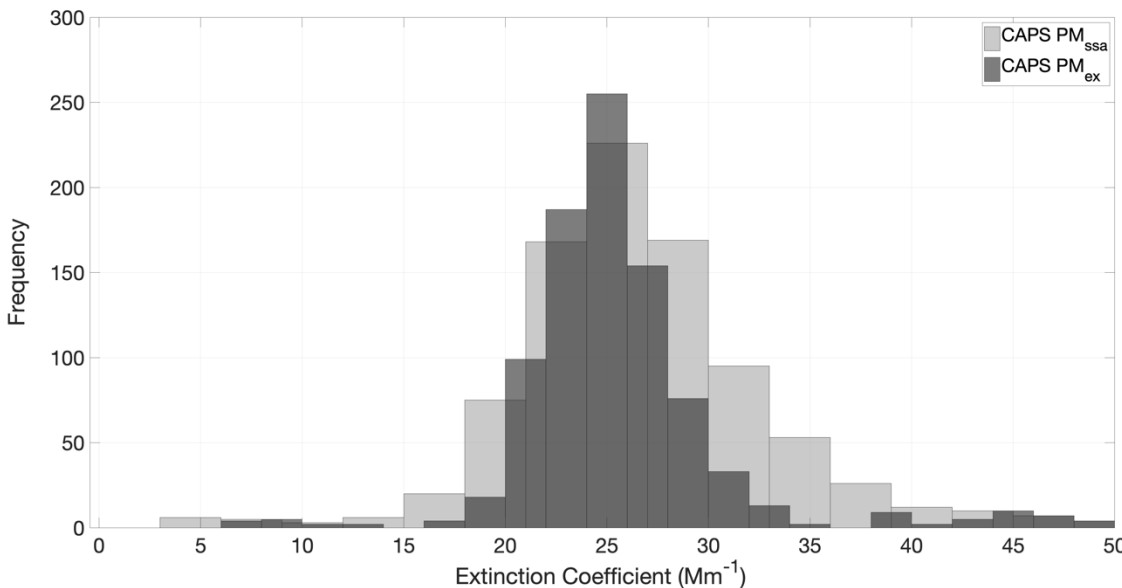

**Figure 7. Frequency of extinction coefficient measurement for the CAPS PM_ssa and PM_ex systems at the nominal 25 Mm$^{-1}$ (level 5) test point for AD.**

### 3.2    Scattering Coefficient

The scattering channel of the CAPS PM_ssa was evaluated in comparison to the NEPH measurements for AD, BC, AS, and MIX (Table 1). The time series of scattering coefficient data for the various aerosol runs is shown in Figure 8. Supplemental Table A2 shows the average and 1-σ standard deviation obtained for the targeted scattering coefficient levels. Within the reported error margins of the two instruments we could not observe a systematic deviation of both either in the average or in the standard deviation of the measured values. The precision of both instruments for the measurement of scattering coefficient is very similar.

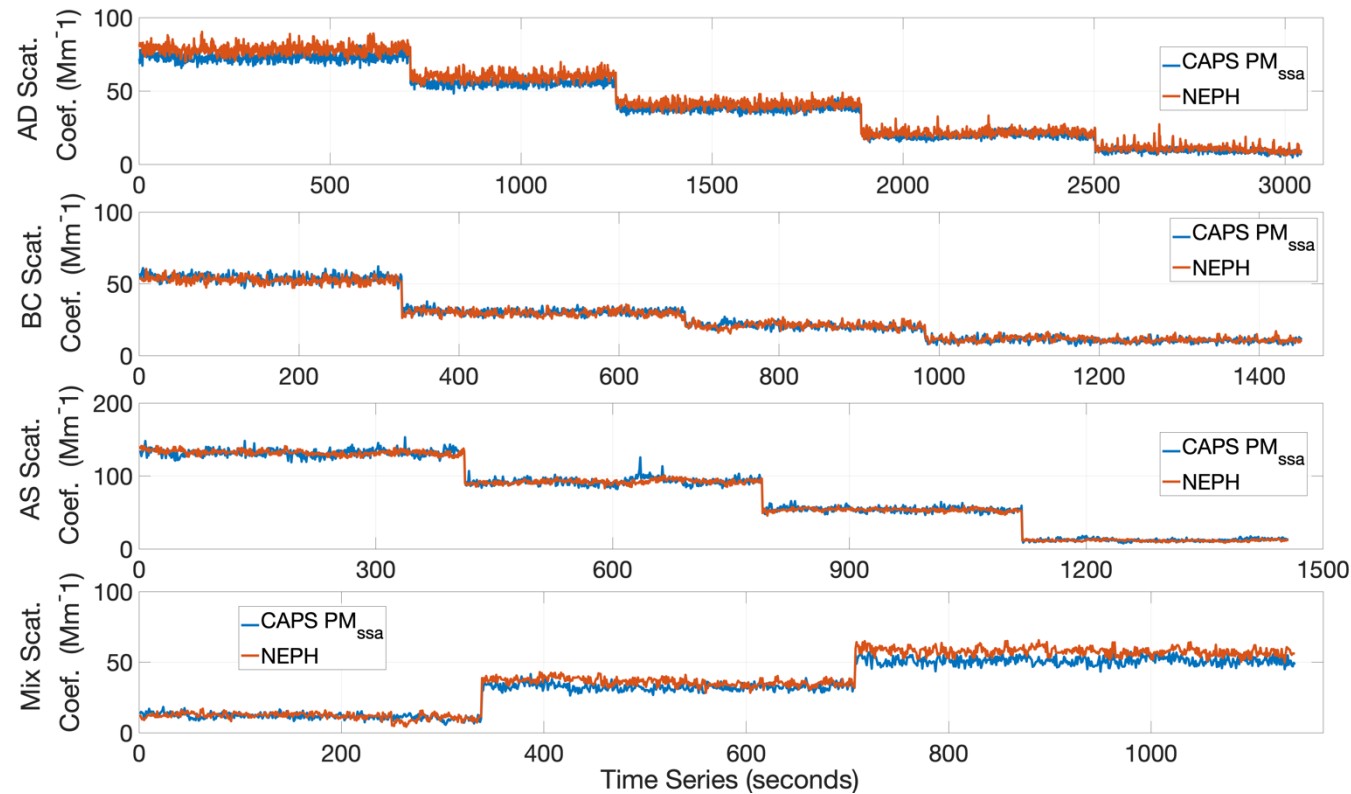

**Figure 8. Time series of the scattering coefficients parallel measured by the different instruments: CAPS PM$_{SSA}$ and NEPH for the different aerosol types (BC (top-) AS (middle-) and Mix (bottom-figure).**

Figure 9 shows the scatter plot of the average and standard deviation for each level of the CAPS PM$_{ssa}$ against NEPH. As it can be seen from Figure 9 and the data compiled in Table 8 (regression line values for the 10s average data), the agreement with the NEPH measurements is good, with less than 10% difference in the slope, offset smaller than 1.6 Mm$^{-1}$ and correlation coefficient of minimum 0.97 for all aerosol types. For the AD, BC and Mix cases, the NEPH data were corrected with the Massoli et al. (2009) approach. For the AS case both the Anderson et al. (1998) and Müller et al. (2011) were applied and the results given were practically the same, less than 2% in the slope and less than 1.00 Mm$^{-1}$ difference in the offset.

**Table 8. Linear regression parameters obtained by fitting 10 second averaged data including the slope (M), standard deviation of the slope (Std m), intercept (B), standard deviation of intercept (std b), and linear regression coefficient (R$^2$) for the comparison of the CAPS PM$_{ssa}$ scattering channel with NEPH**

| Aerosol | Reference Instrument | Nominal SSA | m | Std m | b | Std b | R$^2$ |
|---------|---------------------|-------------|------|-------|-------|-------|------|
| AS | NEPH | 1.00 | 0.99 | 0.00 | 1.28 | 0.24 | 0.99 |
| AD | NEPH | 0.40 | 0.94 | 0.00 | -0.52 | 0.05 | 1.00 |
| BC | NEPH | 0.40 | 1.04 | 0.01 | -0.79 | 0.16 | 0.97 |
| MIX | NEPH | 0.60 | 0.91 | 0.01 | 1.50 | 0.11 | 0.99 |

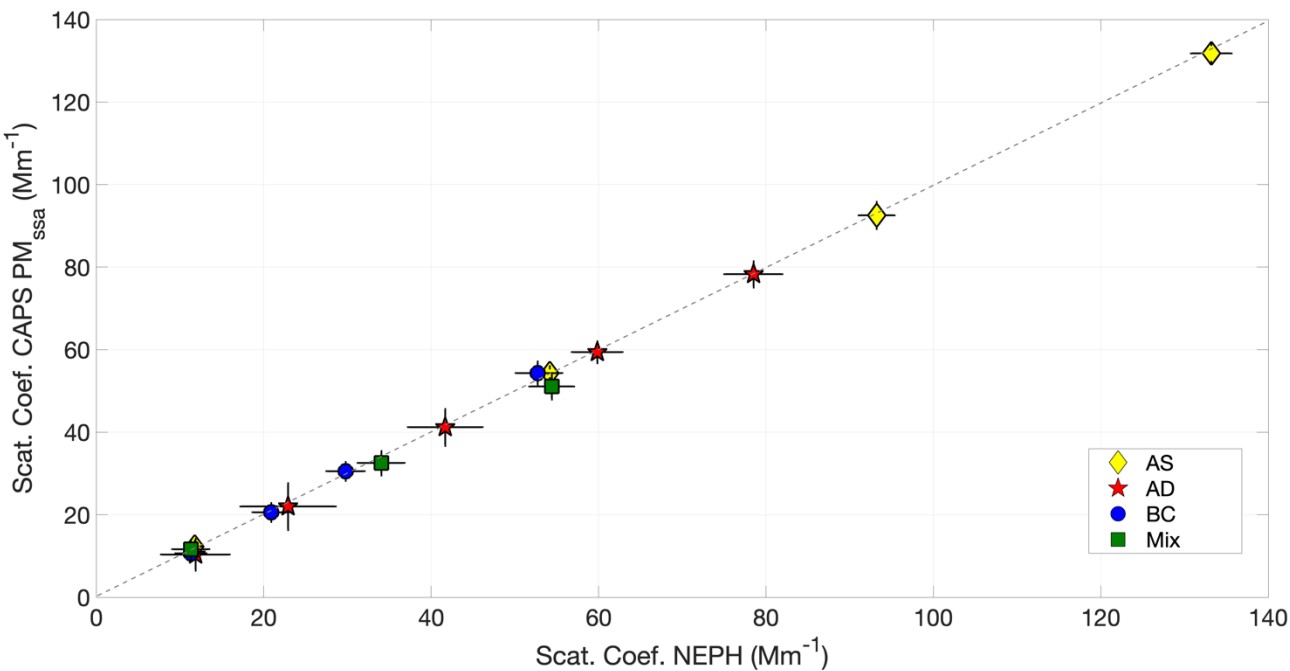

**Figure 9. Comparison result (mean value and standard deviation for each level) of the scattering channel of the CAPS PM$_{ssa}$ with the measurements from the NEPH for the different aerosol types (dashed line is the identity line (i.e., 1:1 line)).**

### 3.3 Absorption Coefficient

In spite of the fact that the CAPS PM$_{ssa}$ is not capable of directly measuring the absorption coefficient, the values can be derived as the difference of the extinction and the scattering coefficients; see Equation (1). From the difference of the two CAPS PM$_{ssa}$ channels the calculated absorption coefficients were compared to the direct measurement by the PSAP. In this analysis, when operating with a mixture of AS and AD, the PSAP data were treated using the correction from Ogren (2010). The time series for the measurement of the different aerosols are shown in Figure 10 whereas Supplemental Table A3 shows the average and 1-σ standard deviation obtained for the targeted absorption coefficient levels.

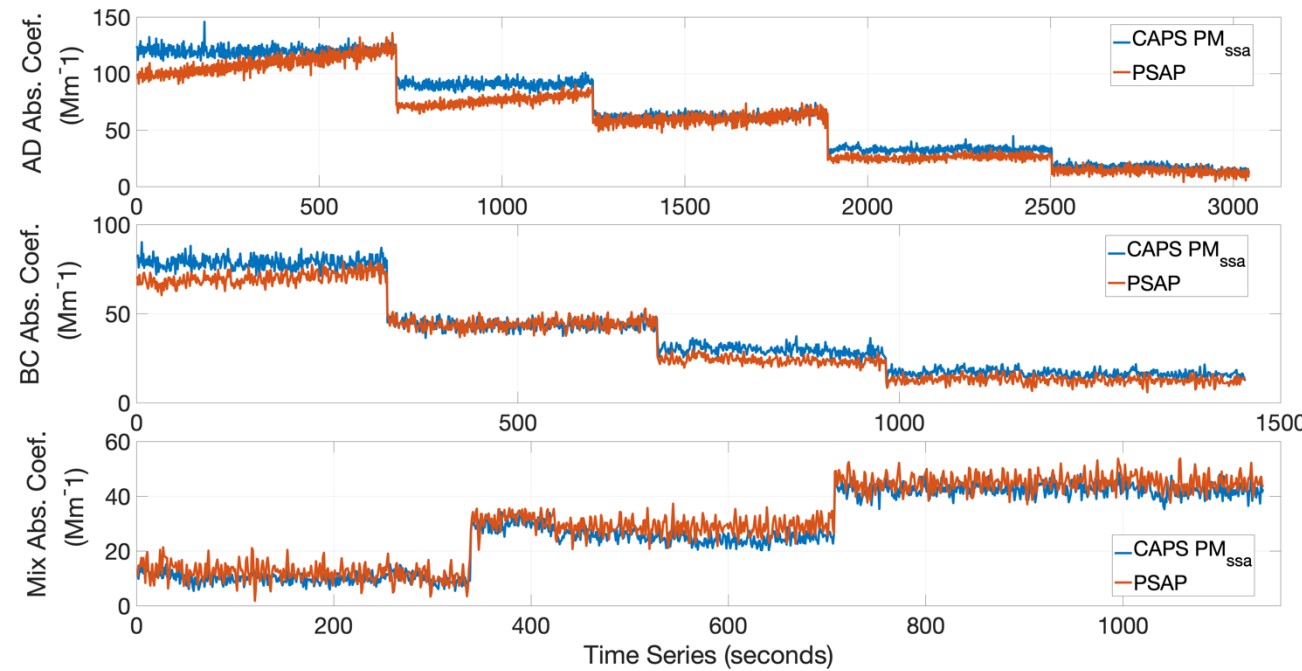

**Figure 10. Time series (in seconds – 1 second resolution) of the parallel measurements of the absorption coefficient for the different test aerosols (TOP (Aquadag AD, Middle BC, Bottom (Mix) by the PSAP and the CAPS PM$_{SSA}$ (as a result of the subtraction of the scattering coefficient from the extinction coefficient).**

The scatter plot for the average measured values from both methods for all levels is shown in Figure 11, whereas the results of the linear regression analysis of the 10 seconds averaged values are compiled in Table 9. The agreement between the methods is reasonable, with deviations below 17% in the slope, and offsets less than 3.0 Mm$^{-1}$. The correlation coefficient is above 0.98 for all cases. Figure 11 demonstrates that for higher absorption coefficients, the two methods deviate more strongly than for lower absorption coefficients. This is mainly caused by the correction algorithm applied to the PSAP data (also seen on Figure 10); filter loading corrections are significantly larger for higher absorption coefficient levels than for lower absorption coefficient levels. The increase in the absorption coefficient observed in Figure 10 for the higher levels of AD and BC, is related to the transmission decay of the filter in the PSAP and the correction algorithm chosen for this study. This finding proves that, although the CAPS PM$_{ssa}$ cannot directly measure aerosol light absorption, it provides a rather reliable measurement of the absorption coefficient of the sampled aerosol, at least for the small particle sizes and intermediate SSA values sampled in this study. The relative accuracy of absorption measurements by the two channels of the CAPS PM$_{ssa}$ may be significantly reduced for weakly absorbing but large-sized and irregularly shaped mineral dust particles.

**Table 9. Linear regression parameters obtained by fitting 10 second averaged data including the slope (M), standard deviation of the slope (Std m), intercept (B), standard deviation of intercept (std b), and linear regression coefficient (R$^2$) for the comparison of the CAPS PM$_{ssa}$ and the PSAP instruments.**

| Aerosol | Reference Instrument | m | Std m | b | Std b | R$^2$ |
|---------|---------------------|------|-------|-------|-------|------|
| AD | PSAP | 1.12 | 0.00 | -2.84 | 0.25 | 0.98 |
| BC | PSAP | 1.04 | 0.00 | 2.68 | 0.16 | 0.98 |
| MIX | PSAP | 1.16 | 0.00 | -2.83 | 0.09 | 0.99 |

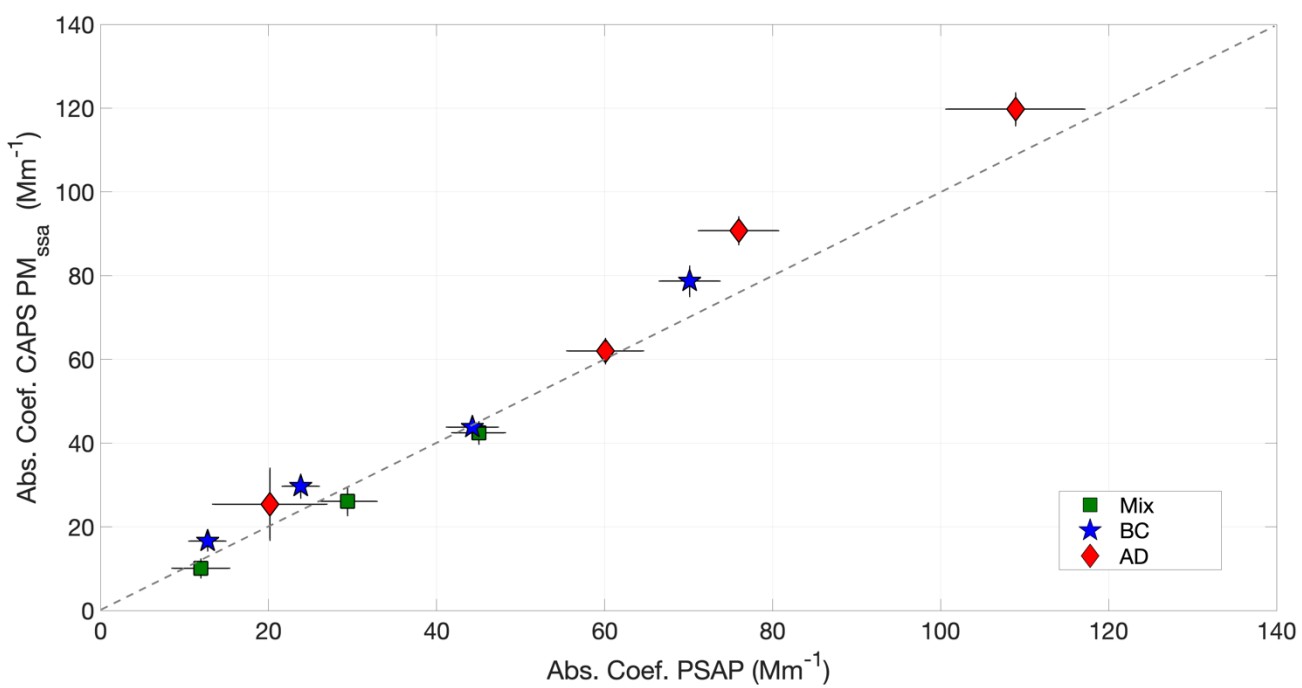

**Figure 11. Comparison result (mean value and standard deviation for each level) of the absorption indirect measurement by the CAPS PM$_{ssa}$ with the measurements from the PSAP for AD, BC and Mixture (dashed line is the identity line (i.e., 1:1 line)).**

**3.4 Single Scattering Albedo Measurement**

The ultimate property targeted by the CAPS PM$_{ssa}$ is the aerosol single-scattering albedo. Figure 12 shows the average and standard deviation of the SSA measured by the CAPS PM$_{ssa}$ and the applied proven technologies for each aerosol type containing a light-absorbing fraction, at the different extinction coefficient levels. The values for each level are also compiled in Supplemental Table A4.

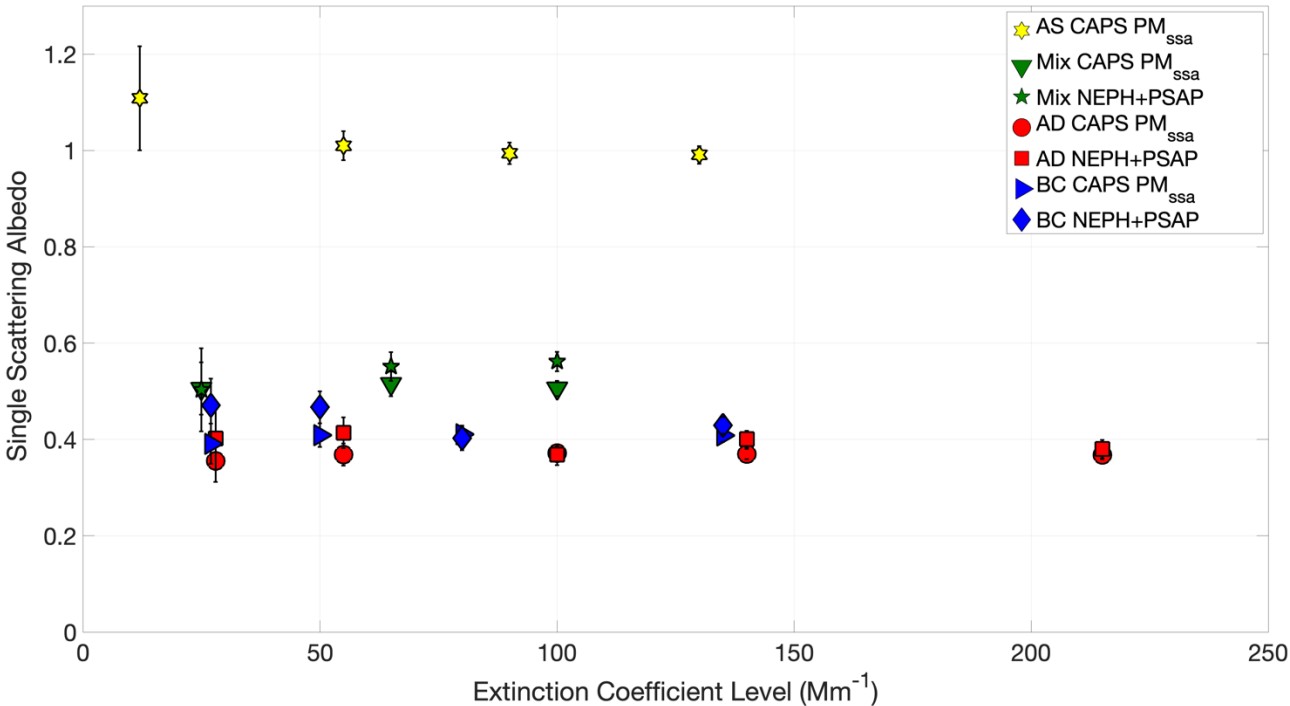

**Figure 12. Average and standard deviation of the measured Single Scattering Albedo as a function of extinction coefficient level for the different aerosols and technologies.**

For the absorbing aerosols, we found maximum deviations between the different SSA values of 0.08, or 17%, with the residuals being randomly distributed around zero. For a single aerosol type, the SSA provided by the CAPS $PM_{ssa}$ shows less scatter around the average value compared to the values derived from PSAP and NEPH data. The measurements by the CAPS $PM_{ssa}$ are more robust in terms of stability in comparison with the values measured by the PSAP+NEPH combination, with an average of the standard deviation for the different aerosol types of 0.025 for the CAPS $PM_{ssa}$ and 0.035 for the PSAP-NEPH combination. It is worth noting that even though there are differences found in the measurements, all measured SSA values fall within the range of values expected for each aerosol type (as measured and detailed in section 3.2 - Table 8).

## 4    Summary and Outlook

An optical closure study has been performed using different types of aerosols (pure scattering, strongly absorbing, and mixture) to evaluate the performance and relative accuracy of the recently launched Cavity Attenuated Phase-Shift Single Scattering Albedo Monitor.

The results from the instrument intercomparison with proven technologies (CAPS $PM_{ex}$, NEPH, and PSAP) show a very good agreement for all aerosol types, with relative accuracy of minimum 90%, for all aerosol types. The small deviation observed in the extinction channel between the CAPS $PM_{ssa}$ and PSAP-NEPH combination originates from the applied correction algorithm to the PSAP data, since it is a logarithmic function of the filter transmission leading to deviations in the dataset. For the evaluation of the performance for each aerosol individually, the extinction channel shows relative accuracy between 94% and 99%; and the scattering channel, between 91% and 99%. These values are very similar to those found by Petzold et al. (2013) for the CAPS $PM_{ex}$.

Regarding the application of the CAPS PM$_{ssa}$ for the measurement of the absorption coefficient and single-scattering albedo, the instrument has shown good performance for the SSA measurement. but only reasonable performance for the absorption. The relative accuracy of the absorption coefficient measurement by the CAPS PM$_{ssa}$ in comparison with the PSAP between 84 and 96% as obtained for the linear regression analysis for all investigated aerosol types and aerosol loadings. The large difference observed here comes from the correction scheme applied to the PSAP data at high loadings, as stated earlier. It is possible to observe that the higher deviations occur at high absorption coefficient, also where the transmission of the filter has a steeper decrease. For the measurement of SSA, the CAPS PM$_{ssa}$ showed a very good stability for all measured $\sigma_{ep}$ levels, better than the PSAP-NEPH combination. The measured values are within what is expected for the different types of aerosols (0.4 for strongly absorbing aerosols and 1.0 for purely scattering aerosols).

The results reported from our study demonstrate that the CAPS PMssa is a very robust and reliable instrument for the direct measurement of the scattering and extinction coefficient, as well as for the indirect measurement of the absorption coefficient and single scattering albedo within the expected limits reported by the error propagation analysis.

## 5   Author Contributions

JP, UB, and AP designed the study and prepared the manuscript, with contributions from all co-authors. AF and TO provided technical details of the instrumentation and contributed to the interpretation of the study results.

## 6   Competing Interests

The authors declare that they have no conflict of interest.

## 7   Acknowledgements

Parts of this work was funded by the EU FP7 project IGAS (Grant Agreement No. 312311), the Federal Ministry of Education and Research, Germany, in IAGOS D (Grant Agreement No. 01LK1301A), EU H2020 Project ENVRIplus (Grant No. 654182) and HITEC Graduate School for Energy and Climate.

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

**9   Appendix A**

ERROR PROPAGATION FOR DERIVED PROPERTIES

**A1   Error propagation for the single scattering albedo using scattering and absorption:**

Error propagation for the single scattering albedo $\omega$ using independent scattering- ($\sigma_{sc}$) and absorption-coefficient ($\sigma_a$) measurements is given by:

$$\omega = \frac{\sigma_{sc}}{\sigma_{sc+}\sigma_a}$$

$$\Delta\omega = \sqrt{\underbrace{\left(\frac{\partial\omega}{\partial\sigma_{sc}} * \Delta\sigma_{sc}\right)^2}_{1} + \underbrace{\left(\frac{\partial\omega}{\partial\sigma_a} * \Delta\sigma_a\right)^2}_{2}}$$

Using

$$1) f(x) = \frac{x}{x + a} \quad \rightarrow f'(x) = \frac{a}{(x + a)^2}$$

$$2) g(x) = \frac{b}{b + x} \quad \rightarrow g'(x) = \frac{-b}{(x + b)^2}$$

We end up:

$$\Delta\omega = \sqrt{\left(\frac{\sigma_{sc}}{(\sigma_{sc}+\sigma_a)^2} * \Delta\sigma_{sc}\right)^2 + \left(\frac{-\sigma_a}{(\sigma_{sc}+\sigma_a)^2} * \Delta\sigma_a\right)^2} \tag{1}$$

**A2   Error propagation for the single scattering albedo using scattering and extinction:**

Error propagation for the calculated single scattering albedo using CAPS_ssa measurements of scattering- ($\sigma_{sc}$) and extinction- coefficients ($\sigma_e$).

$$\omega = \frac{\sigma_{sc}}{\sigma_e}$$

$$\Delta\omega = \sqrt{\left(\frac{\partial\omega}{\partial\sigma_{sc}} * \Delta\sigma_{sc}\right)^2 + \left(\frac{\partial\omega}{\partial\sigma_e} * \Delta\sigma_e\right)^2}$$

$$\Delta\omega = \sqrt{\left(\frac{1}{\sigma_e} * \Delta\sigma_{sc}\right)^2 + \left(\frac{-\sigma_{sc}}{\sigma_e^2} * \Delta\sigma_e\right)^2} \tag{2}$$

The error propagation of the calculated absorption coefficient using CAPS_ssa measurement (independent scattering and extinction measurements) is given by:

$$\sigma_a = \sigma_e - \sigma_{sc}$$

$$\Delta\sigma_a = \sqrt{\left(\frac{\partial\sigma_a}{\partial\sigma_{sc}} * \Delta\sigma_{sc}\right)^2 + \left(\frac{\partial\sigma_a}{\partial\sigma_e} * \Delta\sigma_e\right)^2}$$

$$\Delta\sigma_a = \sqrt{(-\Delta\sigma_{sc})^2 + (\Delta\sigma_e)^2}$$

**General remark:**

The error of a mean value using n values of x is given by:

$$x_{mean} = \sum_{i=1}^{n} \frac{x_i}{n}$$

$$\Delta x_{mean} = \frac{\Delta x}{\sqrt{n}}$$

In the following graphs relative errors are reported defined by

510 $$\text{rel. error }(x) = \frac{\Delta x}{x}$$

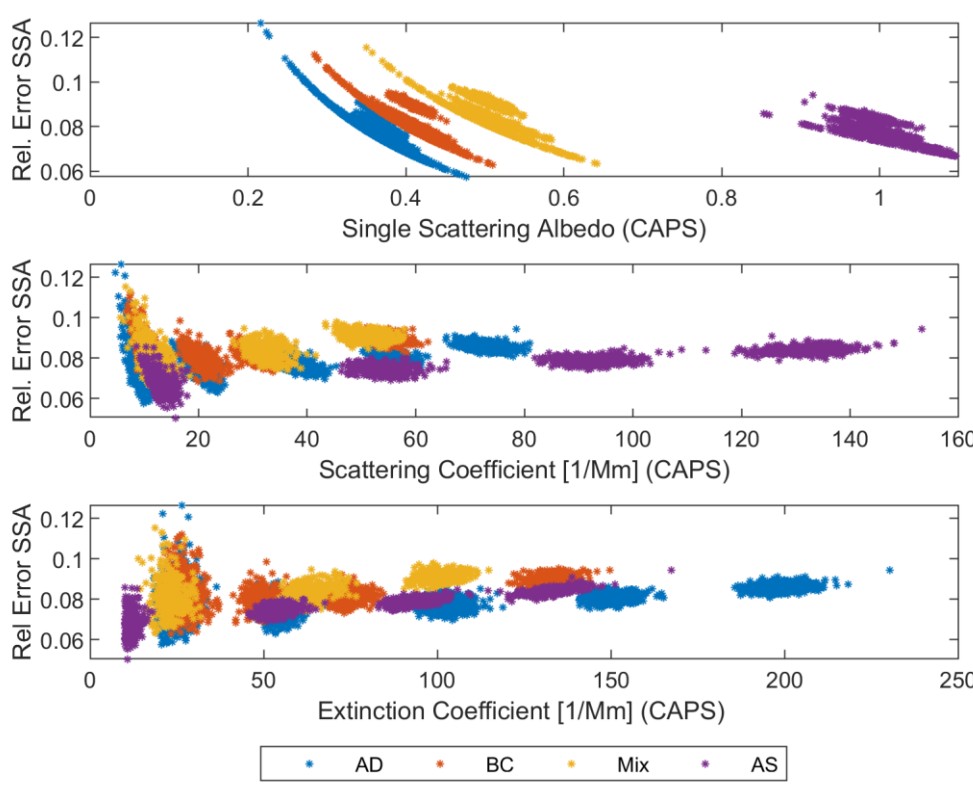

**Figure 13 Relative error of the Single Scattering Albedo (SSA) derived from CAPS measurements as function of the derived SSA**
**(top-), the scattering coefficient (middle-), the extinction coefficient (base-plot).**

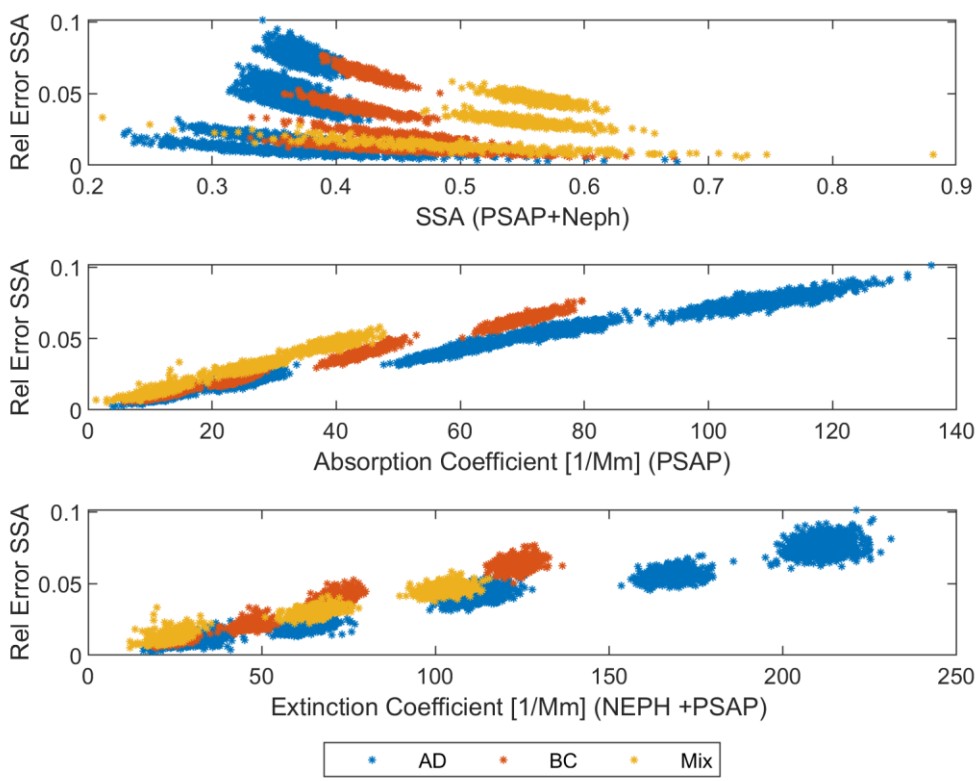

**Figure 14 Relative error of the single scattering albedo derived from PSAP and TSI nephelometer data as function of the SSA (top-), absorption coefficient (middle-), extinction coefficient (base-plot).**

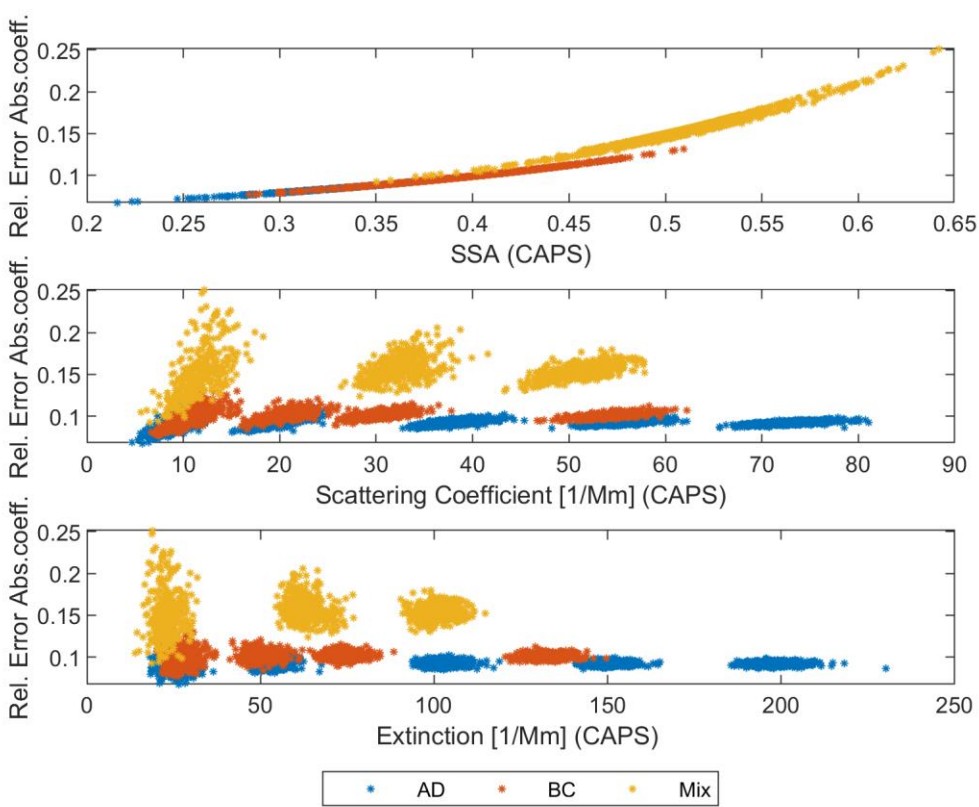

**Figure 15 Relative error of the absorption coefficients using CAPS measurements as function of SSA (top-), scattering coefficient (middle-), and extinction coefficient (base-plot).**

## 10 Appendix B

**Table A1.** Extinction coefficient mean and 1-σ standard deviation of the mean measured by the CAPS PM$_{ssa}$ extinction channel and proven technologies

|     |             |     | Run 1 | Run 2 | Run 3 | Run 4 | Run 5 |
|-----|-------------|-----|-------|-------|-------|-------|-------|
| AS  | CAPS PM$_{ssa}$ | Av  | 54.62 | 127.43 | 311.65 | 198.31 | NA |
|     |             | Std | 15.94 | 17.48 | 41.38 | 37.91 | NA |
|     | CAPS PM$_{ex}$ | Av  | 53.39 | 124.78 | 306.40 | 195.94 | NA |
|     |             | Std | 11.41 | 10.76 | 27.04 | 25.55 | NA |
| AD  | CAPS PM$_{ssa}$ | Av  | 221.04 | 105.98 | 66.16 | 26.25 | 8.84 |
|     |             | Std | 28.85 | 6.04 | 6.00 | 5.75 | 3.35 |
|     | CAPS PM$_{ex}$ | Av  | 210.15 | 100.22 | 63.08 | 24.93 | 8.66 |
|     |             | Std | 33.05 | 5.64 | 4.39 | 4.03 | 2.08 |
| AD  | CAPS PM$_{ssa}$ | Av  | 198.00 | 150.09 | 104.15 | 56.88 | 28.85 |
|     |             | Std | 5.32 | 4.02 | 9.83 | 13.02 | 8.53 |
|     | PSAP+NEPH   | Av  | 187.37 | 135.55 | 102.30 | 51.34 | 26.78 |
|     |             | Std | 8.94 | 5.45 | 9.00 | 15.00 | 8.00 |
| BC  | CAPS PM$_{ssa}$ | Av  | 136.77 | 76.16 | 50.99 | 27.73 | NA |
|     |             | Std | 1.46 | 1.36 | 3.03 | 1.61 | NA |
|     | PSAP+NEPH   | Av  | 134.98 | 81.59 | 48.51 | 26.28 | NA |
|     |             | Std | 2.02 | 1.28 | 1.95 | 1.34 | NA |
| Mix | CAPS PM$_{ssa}$ | Av  | 23.05 | 63.14 | 100.94 | NA | NA |
|     |             | Std | 3.06 | 4.88 | 4.20 | NA | NA |
|     | PSAP+NEPH   | Av  | 23.23 | 63.36 | 99.47 | NA | NA |
|     |             | Std | 4.20 | 4.37 | 4.51 | NA | NA |

**Table A2.** Scattering coefficient mean and 1-σ standard deviation of the mean measured by the CAPS $PM_{ssa}$ and NEPH

|  |  |  | Run 1 | Run 2 | Run 3 | Run 4 | Run 5 |
|---|---|---|---|---|---|---|---|
| AS | CAPS $PM_{ssa}$ | Av | 131.79 | 92.57 | 54.29 | 12.31 | NA |
|  |  | Std | 2.26 | 3.02 | 1.43 | 1.18 | NA |
|  | NEPH | Av | 133.22 | 93.22 | 54.18 | 11.77 | NA |
|  |  | Std | 2.29 | 2.03 | 1.36 | 0.82 | NA |
| AD | CAPS $PM_{ssa}$ | Av | 78.29 | 59.42 | 41.18 | 21.98 | 10.32 |
|  |  | Std | 2.89 | 2.37 | 4.18 | 5.44 | 3.59 |
|  | NEPH | Av | 78.50 | 59.86 | 41.70 | 22.93 | 11.87 |
|  |  | Std | 3.33 | 2.90 | 4.31 | 5.56 | 3.98 |
| BC | CAPS $PM_{ssa}$ | Av | 54.33 | 30.54 | 20.58 | 10.66 | NA |
|  |  | Std | 2.57 | 2.03 | 1.99 | 1.70 | NA |
|  | NEPH | Av | 52.71 | 29.81 | 20.91 | 11.31 | NA |
|  |  | Std | 2.46 | 2.15 | 2.09 | 1.71 | NA |
| Mix | CAPS $PM_{ssa}$ | Av | 11.66 | 32.52 | 51.09 | NA | NA |
|  |  | Std | 1.98 | 2.66 | 2.83 | NA | NA |
|  | NEPH | Av | 11.32 | 34.05 | 54.43 | NA | NA |
|  |  | Std | 2.09 | 2.67 | 2.55 | NA | NA |

**Table A3.** Absorption coefficient mean and standard deviation of the mean measured by the CAPS $PM_{ssa}$ (extinction minus scattering) and PSAP

|  |  |  | Run 1 | Run 2 | Run 3 | Run 4 | Run 5 |
|---|---|---|---|---|---|---|---|
| BC | CAPS $PM_{ssa}$ | Av | 78.69 | 43.78 | 29.73 | 16.57 | NA |
|  |  | Std | 3.28 | 2.53 | 2.50 | 1.94 | NA |
|  | PSAP | Av | 71.11 | 41.45 | 25.86 | 13.35 | NA |
|  |  | Std | 2.83 | 2.56 | 2.27 | 2.10 | NA |
| AD | CAPS $PM_{ssa}$ | Av | 119.75 | 90.76 | 62.02 | 32.87 | 16.93 |
|  |  | Std | 3.55 | 2.99 | 2.69 | 2.10 | 2.33 |
|  | PSAP | Av | 133.00 | 108.34 | 71.50 | 41.10 | 19.83 |
|  |  | Std | 4.69 | 3.69 | 4.22 | 3.38 | 3.91 |
| Mix | CAPS $PM_{ssa}$ | Av | 10.09 | 26.04 | 42.45 | NA | NA |
|  |  | Std | 1.88 | 2.85 | 2.37 | NA | NA |
|  | PSAP | Av | 11.95 | 29.37 | 45.04 | NA | NA |
|  |  | Std | 3.27 | 3.17 | 3.02 | NA | NA |

**Table A4.** Single Scattering Albedo average value and standard deviation for CAPS PM$_{ssa}$ and proven technologies

| | Scat/Ext | | Run 1 | Run 2 | Run 3 | Run 4 | Run 5 |
|---|---|---|---|---|---|---|---|
| AS | CAPS PM$_{ssa}$ | Av | 0.99 | 0.99 | 1.01 | 1.11 | NA |
| | | Std | 0.02 | 0.02 | 0.03 | 0.11 | NA |
| AD | CAPS PM$_{ssa}$ | Av | 0.37 | 0.37 | 0.37 | 0.37 | 0.36 |
| | | Std | 0.01 | 0.01 | 0.01 | 0.02 | 0.04 |
| | PSAP+NEPH | Av | 0.38 | 0.40 | 0.37 | 0.41 | 0.40 |
| | | Std | 0.02 | 0.02 | 0.02 | 0.03 | 0.07 |
| BC | CAPS PM$_{ssa}$ | Av | 0.41 | 0.41 | 0.41 | 0.39 | NA |
| | | Std | 0.01 | 0.02 | 0.02 | 0.04 | NA |
| | PSAP+NEPH | Av | 0.43 | 0.40 | 0.47 | 0.47 | NA |
| | | Std | 0.02 | 0.02 | 0.03 | 0.05 | NA |
| Mix | CAPS PM$_{ssa}$ | Av | 0.51 | 0.52 | 0.51 | NA | NA |
| | | Std | 0.05 | 0.03 | 0.02 | NA | NA |
| | PSAP+NEPH | Av | 0.50 | 0.55 | 0.56 | NA | NA |
| | | Std | 0.08 | 0.03 | 0.02 | NA | NA |