# Peer review of "Laboratory Validation of a Compact Single-Scattering Albedo (SSA) Monitor"

_Atmospheric Measurement Techniques, 2019_

## Referee Comment (RC2)

REVIEW NOTES and comments:

**Laboratory Validation and Field Deployment of a Compact Single-Scattering (SSA) Albedo Monitor**

Journal: AMT
Title: Laboratory Validation and Field Deployment of a Compact Single-Scattering Albedo (SSA) Monitor
Author(s): Julia Perim de Faria et al.
MS No.: amt-2019-146
MS Type: Research article

Matrix Scores:   Criteria:          **Scientific Significance – Good 2**

**Scientific Quality – Fair 4**

**Presentation Quality - Fair**

Over all English language presentation:

There is a general non-standard usage of comma separators, and a few general awkward English syntax constructions.  However, it seems most intended meanings are clear.  The paper could use revision of grammatical and syntactical usage to make the reading flow more smoothly.

Over all Scientific Presentation:

General lack of the definition and standard used for the terms accuracy and precision.  There should be at least an equation presented for the calculation assumed in each measure.  It is important as the system of closure for the complete instrumental experimental circuit depends not only upon the accuracy and precision of each individual instrument, but the data path through all of them.

The study could be enhanced by a true presentation of error propagation by classical form differential error analysis.  The assumption of normally [Gaussian} distributed error seems perhaps unfounded in such a complicated closure strategy.

The work has merit and should be published conditional upon appropriate revisions and additions.

Specifics:

Table1:  the mixture of AS+AD is assumed to have an SSA $_{\lambda=630nm}$ of 0.6 for the study case, but lacks details in discussion of how the mixture of the standard substances was to be controlled.

Lines 100 – 105:

Perhaps some calibration data could be presented, as well as plot of Scattering Channel signal vs. Extinction Channel signal.  This could provide insight into baseline fluctuations and possible instrumental bias.

Section 2.1.2

Lines 115 -120  "The instrument measures....."

Section 2.1.3  CAPS PMext configuration

It might be beneficial to include a figure as nicely detailed as that of Figure 2. For the CAPS PMssa configuration.

Lines 140-145

One of the unique features of the CAPS PMssa set-up is the integrating sphere.  The glass tube that passes through the sphere needs a bit more detailed information as it is inside the integrator.  Some specifics as to the thickness of the wall, any coating it may have, it's optical properties should be characterized or listed somewhere from the manufacturer or supplier - if not determined during calibration of the instrument itself.

[A general Question:  Were any other wavelengths considered or tested for the calibration standard?]

Section 3 discussion:

Some of the sentences could be divided into shorter more clear constructions

Lines 210 – 215:

1) I think these critical figures could be sized up a bit
2) There seems to be a general assumption that the standard deviation is the most reliable measure of experimental uncertainty.  This reviewer is not sure this is a completely valid assumption.
3) The reference to PSAP-NEPH extinction measurements being similar to those of Petzold 2013: this unfortunately requires the reader to find the other paper to validate this statement of event or know what the expected result was. A simple sentence could clarify this. [Yes, as one of the contributing authors it is perfectly acceptable to cite their own previous research articles, but perhaps a bit much to expect the reader of this article to be familiar with the result of that work.]

Section 3.2

Line 217:  " There is no systematic error found neither in the average nor in the standard deviation of the measured values."  Although the internal reference is to a table included in the supplemental material, it is a mathematically unsupported assertion.  A calculation or insight into how this statement is evidenced might make a stronger case for its inclusion.

Section 3.3

Figure10: is problematic on multiple levels: although the notion of overlying *timeseries* into a single track representing the CAPS PMssa and PSAP for the three type of aerosol particles testing is a good idea, the diagram has flaws. [same comments apply to Figure 7 on the scattering channe;]

1) The figures do not expand into full size charts and are presented TOO SMALL to intuit any scientific sense from their visual examination. [This may be a display result after the Copernicus online system was revamped for their paper display] But the authors could simple make a much larger figure.
2) The horizontal axis has numbers on a scale with no mention or label as to their units. Are they "seconds" after the calibration sequence has finished? Are they minutes?
3) Even if the individual axis numbers align, there is not a mention to assure the reader they were simultaneously measured.
4) These figures as a set need to be amplified in the vertical scale so as to make visible Any regions in time where the CAPS PMssa signal fluctuations and spikes might not be synchronous to those of the supposed time coincident signal of the PSAP.
5) Expanding the horizontal time axes will allow the reader to view regions where the signals might not be precisely time correlated and any instrumental fluctuations as "noise."

Discussion of this diagram is not complete.  No mention is made to the significance of the regions where the traces converge over time to a common point in the AD and BC examples.  No mention of the significance, if any, of time intervals where the signals step down, or step-up in sigma ($\sigma$).

NOTE: as mentioned prior it is not sufficient to cite a method "data correction" (Ogren 2010) without explaining why it is appropriate in this situation and how it fundamentally treats the data.  Forcing the reader to find another paper to understand what is going on in this paper is not exercising good scientific communication skill.  There is nothing wrong with the citation of Ogren 2010, simply the authors here should explain how and why is it used, as well as it's importance to the data collected in this research.

It should also be noted that without a time series analysis proper [lag correlation, etc. as an example] there is not a reliable method to indicate how the static correlation coefficients presented in the table evolve over time as the instruments run.  Correlation coefficients are important as measures, but should state clearly they might not reveal complex interrelationships between data signals as the instruments run over time.

Section 3.4

This is the key portion of the research and should be strongly emphasized. Generally well done.

Line 281:  does the statement "....expected values for each aerosol type" directly refer to table 1?

If so, reiterate that.   If not, please summarize the expected values directly here.

THIS IS ENOUGH TO WORK ON FOR now

---

## Referee Comment (RC1) · Anonymous Referee #1 · 29 May 2019

This manuscript provides an analysis of a set of laboratory experiments comparing the recently developed CAPS PMssa instrument, which measures aerosol extinction and absorption (from which is derived single-scatter albedo) using cavity phase-shift and integrating sphere techniques, respectively. Because this instrument can determine SSA through from a single sample, and bypasses the need for relatively uncertain absorption measurements using filter media, it has the potential to be extremely valuable. The aerosol direct effect remains a large uncertainty in the Earth's radiative balance, and this instrument, if accurate and widely used, has the potential to help reduce this uncertainty. The topic is entirely within the scope of AMT and there should be many readers interested in the subject. The experiments described in the manuscript appear

to be well conducted and have produced high quality data.

Regrettably, there are some significant issues with the manuscript, two especially, that will require a major revision. These two issues are:

1) There is no error analysis of the techniques being compared. Instead, variance in the measurements is used as surrogate for uncertainty. The authors need to directly and independently determine the uncertainty in the phase-shift extinction measurement and the uncertainty in the scattering measurement, and propagate these uncertainties through to the final SSA product. This uncertainty analysis must include consideration of both potential biases (which might include determination of pressure and temperature in the instrument, for example) and random uncertainties (which might include noise in the measurement that requires averaging). As it currently stands with this manuscript, if I were to purchase two of the CAPS PMssa instruments and compare them and find that they disagree by 5% in extinction and/or scattering, I would not know if this is within expectations or would indicate a problem with one of the instruments. A complete uncertainty analysis needs to be applied to all the instrument combinations used. In Table 6, it is not at all clear where these "uncertainties" come from; they appear to be either the scatter in the data (plotted in Fig. 12) or else the difference between the mean values of the data and the "expected" values of the calibration aerosols. Because the SSA of the calibration aerosols is not truly known, there is no absolute standard provided against which to evaluate the different approaches to measuring SSA, so a fundamental uncertainty propagation is needed.

2) The linear regressions which provide the bases for the evaluated values appear to have some sort of error. Examining Fig. 4, the fitted line shown on the scatterplot lies below all of the datapoints. Using Table A1, I plotted the data shown in Fig. 4 and performed a linear regression. I got a slope of ∼1.08 and the fitted line passed directly through the data. This compares with the "all" fit shown in Table 3 of the manuscript, which gives a slope of 0.97. I tried a two-sided regression, a one-sided regression, and one-sided regression forced through zero intercept. All gave slopes >1.06. Inspecting

the other scatterplots in the manuscript (e.g. Figs. 8, Fig. 11 for absorptions <70 Mm-1), the fitted slopes do not seem to go through the data. Unless I've made an error, it appears that the values appearing in all the tables are suspect because of this fitting issue. Thank you for including all the data in the supplemental material tables, which makes finding an apparent problem like this easier.

In addition to these two principal issues, there are some smaller items that need addressing.

a) The table captions all need to be more precise. For example, Table 3 might have a caption of, "Linear regression parameters slope (m), intercept (b) and their standard deviations and the linear regression coefficient R2."

b) I was trying to understand for quite some time how the column labeled "SSA" in Table 3 was calculated before realizing that it is simply an estimate of the SSA for the aerosol type being generated. It might be clearer to move the SSA column to the second column of the table and label it "estimated SSA".

c) The figures all need to use a heavier line width and larger, denser font. It is quite hard to read the labels and identify the symbols and lines.

d) The descriptions in the tables and figures of "PSAP-Neph" is confusing; it suggest that you are subtracting the scattering data from the absorption data. I suggest you use PSAP+Neph for Table 1 and Fig. 4, and "PSAP & Neph" for Table 6.

e) Figs. 5 and 9 are not needed since the data appear in tables already.

f) Fig. 12 should also show the SSA determined for the ammonium sulfate aerosol; this would give a good idea of the scatter about a known, non-absorbing compound.

g) The title includes "Field Deployment". There is no field deployment of the instrument described in this manuscript, just laboratory tests.

h) Lines 93 to 97, the description of the roles of MFC#1 and #2 in regulating make-up

air appear to be switched with MFC#3 and #4 in Fig. 1.

i) Section 2, please describe the truncation angles for the various instruments and typical magnitudes of the correction factors. The uncertainty in these correction factors need to be part of the total uncertainty analysis and error propagation.

j) Please use 2-sided (orthogonal distance) regression when performing the linear regressions. There are uncertainties in both x and y dimensions that should be accounted for. Please weight the regressions by the uncertainty in the measurements.

k) The figure captions (e.g., Fig. 3: "Time series of the measurements by the extinction channel" do not adequately describe the contents of the figures, which in this case shows results from 3 different instruments/combinations, not just the extinction channel of the CAPS PMssa. The same for the other time plots.

There is a lot of good information from some carefully performed and important laboratory studies in the manuscript. I encourage the authors to address the concerns indicated above and submit a revised manuscript that more fundamentally addresses uncertainties and that uses accurately determined regression slopes using properly weighted 2-sided linear regressions.

———————————————————

---

## Author Comment (AC1) · 28 Nov 2019

**Answers to the Reviewer#1**

**We thank the anonymous reviewer #1 for the detailed review. Including the suggestions made has significantly enhanced the paper. In the following you will find our response to the reviewer directly marked in red.**

**Anonymous Referee #1**

This manuscript provides an analysis of a set of laboratory experiments comparing the recently developed CAPS PMssa instrument, which measures aerosol extinction and absorption (from which is derived single-scatter albedo) using cavity phase-shift and integrating sphere techniques, respectively. Because this instrument can determine SSA through from a single sample, and bypasses the need for relatively uncertain ab- sorption measurements using filter media, it has the potential to be extremely valuable. The aerosol direct effect remains a large uncertainty in the Earth's radiative balance, and this instrument, if accurate and widely used, has the potential to help reduce this uncertainty. The topic is entirely within the scope of AMT and there should be many readers interested in the subject. The experiments described in the manuscript appear to be well conducted and have produced high quality data.

Regrettably, there are some significant issues with the manuscript, two especially, that will require a major revision. These two issues are:

1) There is no error analysis of the techniques being compared. Instead, variance in the measurements is used as surrogate for uncertainty. The authors need to directly and independently determine the uncertainty in the phase-shift extinction measurement and the uncertainty in the scattering measurement, and propagate these uncertainties through to the final SSA product. This uncertainty analysis must include consideration of both potential biases (which might include determination of pressure and temperature in the instrument, for example) and random uncertainties (which might include noise in the measurement that requires averaging). As it currently stands with this manuscript, if I were to purchase two of the CAPS PMssa instruments and compare them and find that they disagree by 5% in extinction and/or scattering, I would not know if this is within expectations or would indicate a problem with one of the instruments. A complete uncertainty analysis needs to be applied to all the instrument combinations used. In Table 6, it is not at all clear where these "uncertainties" come from; they appear to be either the scatter in the data (plotted in Fig. 12) or else the difference between the mean values of the data and the "expected" values of the calibration aerosols. Because the SSA of the calibration aerosols is not truly known, there is no absolute standard provided against which to evaluate the different approaches to measuring SSA, so a fundamental uncertainty propagation is needed.

This paper does NOT address in any explicit way, nor was it designed to address, the question of the absolute uncertainties of these techniques. It was designed to address the question of how well they correlate. Thus, the results are given in correlation coefficients (slope and intercept) and their statistical uncertainties

We have added Section 2.3 (new) summarizing the measurement uncertainties reported by the different instrument paper. In particular the uncertainty of the CAPS PMssa was described in detail by Onasch et al. (2013). We have added a passage to the manuscript referencing the detailed error analysis by Onash et al. more pronounced. Temperature and pressure variabilities as potential biases are part of the used measurement error and minimized by regular taking baseline measurements.

2) The linear regressions which provide the bases for the evaluated values appear to have some sort of error. Examining Fig. 4, the fitted line shown on the scatterplot lies below all of the datapoints. Using Table A1, I plotted the data shown in Fig. 4 and performed a linear regression. I got a slope of ~1.08 and the fitted line passed directly through the data. This compares with the "all" fit shown in Table 3 of the manuscript, which gives a slope of 0.97. I tried a two-sided regression, a one-sided regression, and one-sided regression forced through zero intercept. All gave slopes >1.06. Inspecting the other scatterplots in the manuscript (e.g. Figs. 8, Fig. 11 for absorptions <70 Mm-1), the fitted slopes do not seem to go through the data. Unless I've made an error, it appears that the values appearing in all the tables are suspect because of this fitting issue. Thank you for including all the data in the supplemental material tables, which makes finding an apparent problem like this easier.

The line shown in the linear regression was misplaced. The line is actually just a 1-1 line to help readers to evaluate the results in comparison to a perfect correlation (1-1). We have replaced the figures and added the information to the figure caption.

In addition to these two principal issues, there are some smaller items that need addressing.

a) The table captions all need to be more precise. For example, Table 3 might have a caption of, "Linear regression parameters slope (m), intercept (b) and their standard deviations and the linear regression coefficient R2."

We have added your suggestion to the captions.

b) I was trying to understand for quite some time how the column labeled "SSA" in Table 3 was calculated before realizing that it is simply an estimate of the SSA for the aerosol type being generated. It might be clearer to move the SSA column to the second column of the table and label it "estimated SSA".

We have added your suggestion to the table.

c) The figures all need to use a heavier line width and larger, denser font. It is quite hard to read the labels and identify the symbols and lines.

We have improved the figures resolution. But a final version (more readable) of each figure will depend on the layout specified by the AMT. Therefore, we will wait until the final version of the article is given (format wise) to optimize the figures sizes and resolutions.

d) The descriptions in the tables and figures of "PSAP-Neph" is confusing; it suggests that you are subtracting the scattering data from the absorption data. I suggest you use PSAP+Neph for Table 1 and Fig. 4, and "PSAP & Neph" for Table 6.

We have added your suggestion

e) Figs. 5 and 9 are not needed since the data appear in tables already.

It is a visual results representation, which we believe is important to many readers.

f) Fig. 12 should also show the SSA determined for the ammonium sulfate aerosol; this would give a good idea of the scatter about a known, non-absorbing compound.

We have added the AS Data to Figure 14 (new)

g) The title includes "Field Deployment". There is no field deployment of the instrument described in this manuscript, just laboratory tests.

Good point! The field deployment was excluded from this article, thus the title has been modified.

h) Lines 93 to 97, the description of the roles of MFC#1 and #2 in regulating make-up air appear to be switched with MFC#3 and #4 in Fig. 1.

Right. It is corrected (lines 96-97) now.

i) Section 2, please describe the truncation angles for the various instruments and typical magnitudes of the correction factors. The uncertainty in these correction factors need to be part of the total uncertainty analysis and error propagation.

There are many studies about the truncation angles and corrections for the proven technologies (most important and used ones are referred in the article). For the SSA Monitor the information has been added to section 2 (lines 196-204) in the description of the instrument, since it is what is being evaluated in this article. For the other instruments, we included the uncertainty section 2-3 citing the relevant literature in a pronounced part.

j) Please use 2-sided (orthogonal distance) regression when performing the linear regressions. There are uncertainties in both x and y dimensions that should be accounted for. Please weight the regressions by the uncertainty in the measurements.

All uncertainties (standard deviations) are presented for both x and y. Sometimes the values are so small that they are smaller than the data point marker.

k) The figure captions (e.g., Fig. 3: "Time series of the measurements by the extinction channel" do not adequately describe the contents of the figures, which in this case shows results from 3 different instruments/combinations, not just the extinction channel of the CAPS PMssa. The same for the other time plots.

We have updated the captions.

There is a lot of good information from some carefully performed and important lab- oratory studies in the manuscript. I encourage the authors to address the concerns indicated above and submit a revised manuscript that more fundamentally addresses uncertainties and that uses accurately determined regression slopes using properly weighted 2-sided linear regressions.

---

## Author Comment (AC2) · 28 Nov 2019

**Answers to the Reviewer#2**

**We thank the anonymous reviewer#2 for the detailed review. Including the suggestions by the referee has significantly enhanced the paper. In the following you will find our response to the reviewer directly marked in red.**

REVIEW NOTES and comments:

**Laboratory Validation and Field Deployment of a Compact Single-Scattering (SSA) Albedo Monitor**

Journal: AMT
Title: Laboratory Validation and Field Deployment of a Compact Single-Scattering Albedo (SSA) Monitor
Author(s): Julia Perim de Faria et al.
MS No.: amt-2019-146
MS Type: Research article

Matrix Scores: Criteria:

**Scientific Significance – Good 2**

**Scientific Quality – Fair 4**

**Presentation Quality - Fair**

Over all English language presentation:

There is a general non-standard usage of comma separators, and a few general awkward English syntax constructions. However, it seems most intended meanings are clear. The paper could use revision of grammatical and syntactical usage to make the reading flow more smoothly.

Over all Scientific Presentation:

General lack of the definition and standard used for the terms accuracy and precision. There should be at least an equation presented for the calculation assumed in each measure. It is important as the system of closure for the complete instrumental experimental circuit depends not only upon the accuracy and precision of each individual instrument, but the data path through all of them.

The study could be enhanced by a true presentation of error propagation by classical form differential error analysis. The assumption of normally [Gaussian} distributed error seems perhaps unfounded in such a complicated closure strategy.

The work has merit and should be published conditional upon appropriate revisions and additions. Specifics:

Table1: the mixture of AS+AD is assumed to have an SSA λ=630nm of 0.6 for the study case, but lacks details in discussion of how the mixture of the standard substances was to be controlled.

The SSA of the mixture containing AS and AD, was controlled by the online measured SSA measured by the CAPS PM$_{ssa}$. See: Line 80-81 in the revised version.

*…. The SSA of the mixture containing AS and AD, was controlled by the online measured SSA measured by the CAPS PMssa. …*

We also added to the head of the table 1) that this is an expected/estimated value.

Lines 100 – 105:

Perhaps some calibration data could be presented, as well as plot of Scattering Channel signal vs. Extinction Channel signal. This could provide insight into baseline fluctuations and possible instrumental bias.

We have added Figure 4 (new) to the revised version.

[Figure]

*Figure 4. Scatterplot of the correlation of the extinction and scattering channel of the CAPS PMssa before and after the calibration using CO2.*

Section 2.1.2

Lines 115 -120 "The instrument measures….."

We have complemented the section.

Section 2.1.3 CAPS PMext configuration

It might be beneficial to include a figure as nicely detailed as that of Figure 2. For the CAPS PMssa configuration.

Figure 2 (new) has been added – although we still have to check the rights to publish a figure from another journal.

[Figure]

*Figure 2. Overview of the main components and operation principle of the CAPS PMex instrument (Massoli et al., 2010)*

Lines 140-145

One of the unique features of the CAPS PMssa set-up is the integrating sphere. The glass tube that passes through the sphere needs a bit more detailed information as it is inside the integrator. Some specifics as to the thickness of the wall, any coating it may have, it's optical properties should be characterized or listed somewhere from the manufacturer or supplier - if not determined during calibration of the instrument itself.

We have added more information in the lines 185 – 193

*Petzold at al. (2013) showed that this purge flow shortens the measurement path and dilutes the sample and requires a correction factor. As done for the CAPS PM$_{ex}$, a new correction factor was developed, by using monodisperse polystyrene spheres (PSL) of know size, for the CAPS PM$_{ssa}$. Due to the cell geometry, the new correction factor was slightly larger than the one found for the extinction monitor, 1.37 and 1.27, respectively (Onasch et al., 2015). The noise of the instrument, trunction angle and instrument uncertainty have also been studied by Onasch et al. (2015). The values found were all below 1 Mm$^{-1}$ for the noise levels (1$\sigma$, 1s) for all wavelengths. For the case of this particular instrument (630 nm), the truncation correction was determined*

*below 4% for typical ambient conditions. The uncertainty was estimated at ±0.03 for SSA equal to 1 (PSL and ammonium sulphate) and decreases to ±0.01 as the SSA goes down.  (630 nm), the truncation correction was determined below 4% for typical ambient conditions. The uncertainty was estimated at ±0.03 for SSA equal to 1 (PSL and ammonium sulphate) and decreases to ±0.01 as the SSA goes down.*

For more details we have to reference Onasch et al. (2013).

[A general Question: Were any other wavelengths considered or tested for the calibration standard? - No]

Section 3 discussion:
Some of the sentences could be divided into shorter more clear constructions Lines 210 – 215:

1)  I think these critical figures could be sized up a bit

We have improved the resolution of the figures, but the final aspects will be determined by the layout from AMT, thus we will wait until the final version (format wise) is done to work on this issue.

2)  There seems to be a general assumption that the standard deviation is the most reliable measure of experimental uncertainty. This reviewer is not sure this is a completely valid assumption.

The Reviewer is right, as long as the standard deviation relays on the assumption that the statistical population follow a Gausian distribution. We have not proven the assumption by applying a Chi square fitting test, but the assumption is not too bad considering the frequency of occurrence diagram (Figure 8 in the revised version) . We have chosen the standard deviation and not a quartile distance (used as distribution free measure) because most readers are used to- and will ask for it.

3)  The reference to PSAP-NEPH extinction measurements being similar to those of Petzold 2013: this unfortunately requires the reader to find the other paper to validate this statement of event or know what the expected result was. A simple sentence could clarify this. [Yes, as one of the contributing authors it is perfectly acceptable to cite their own previous research articles, but perhaps a bit much to expect the reader of this article to be familiar with the result of that work.]

We have added lines 268-269 :

*…similar to the one found by Petzold et al. (2013), in which an excellent correlation (slope of 0.99) was found for the laboratory comparison between the same instruments using  highly absorbing aerosol, exclusive scattering aerosol and mixtures of both"*

Section 3.2

Line 217: " There is no systematic error found neither in the average nor in the standard deviation of the measured values." Although the internal reference is to a table included in the supplemental material, it

is a mathematically unsupported assertion. A calculation or insight into how this statement is evidenced might make a stronger case for its inclusion.

*The reviewer is right. We have changed the sentence*

*Within the error bars of the two instruments we could not observe a systematic deviation of both either in the average or in the standard deviation of the measured values.*

Section 3.3

Figure10: is problematic on multiple levels: although the notion of overlying *timeseries* into a single track representing the CAPS PMssa and PSAP for the three type of aerosol particles testing is a good idea, the diagram has flaws. [same comments apply to Figure 7 on the scattering channel;]

1)  The figures do not expand into full size charts and are presented TOO SMALL to intuit any scientific sense from their visual examination. [This may be a display result after the Copernicus online system was revamped for their paper display] But the authors could simple make a much larger figure.

*We have improved the resolution of the figures, but the final aspects will be determined by the layout from AMT, thus we will wait until the final version (format wise) is done to work on this issue.*

2)  The horizontal axis has numbers on a scale with no mention or label as to their units. Are they "seconds" after the calibration sequence has finished? Are they minutes?

*The Unit (seconds) has been added to the axis label.*

3)  Even if the individual axis numbers align, there is not a mention to assure the reader they were simultaneously measured.

*We have clarified this in the Figure caption.*

4)  These figures as a set need to be amplified in the vertical scale so as to make visible Any regions in time where the CAPS PMssa signal fluctuations and spikes might not be synchronous to those of the supposed time coincident signal of the PSAP.

*Same answer about the figures.*

5)  Expanding the horizontal time axes will allow the reader to view regions where the signals might not be precisely time correlated and any instrumental fluctuations as "noise."

*This would require an interactive zooming option. Unfortunately this is not supported by AMT.*

Discussion of this diagram is not complete. No mention is made to the significance of the regions where the traces converge over time to a common point in the AD and BC examples. No mention of the significance, if any, of time intervals where the signals step down, or step-up in sigma (σ).

*This increase/decrease is seen in figure 12 for the absorption. The explanation is added:*

*"The increase in the absorption coefficient observed in Figure 12 for the higher levels of AD and BC, is related to the transmission decay of the filter in the PSAP and the correction algorithm chosen for this study."*

NOTE: as mentioned prior it is not sufficient to cite a method "data correction" (Ogren 2010) without explaining why it is appropriate in this situation and how it fundamentally treats the data. Forcing the reader to find another paper to understand what is going on in this paper is not exercising good scientific communication skill. There is nothing wrong with the citation of Ogren 2010, simply the authors here should explain how and why is it used, as well as it's importance to the data collected in this research.

We have added all correction functions to the text for completeness. But we did not motivate them in all the details.

I would like to state that it is good practice to reference a data correction algorithms used to the paper without explaining them in all detail as long as they are commonly rated as best practice. If we would have chosen a new/or exotic algorithm then the referee is right. Scientific papers relay on the referencing system. Otherwise the wheel has to be invented again and again- and articles would be more like textbooks. It is no argument that it is some work for a reader to search for a paper. This is part of our job! On the other hand, reading the original literature gives the original authors the credit they should get.

It should also be noted that without a time series analysis proper [lag correlation, etc. as an example] there is not a reliable method to indicate how the static correlation coefficients presented in the table evolve over time as the instruments run. Correlation coefficients are important as measures, but should state clearly they might not reveal complex interrelationships between data signals as the instruments run over time.

In particular the lag correlation (auto- and cross-correlation) does not help for this kind of lab studies. We did not test the instruments dynamically for transfer- response- or relaxation-times. This would be part of an extra paper. A detailed time series analysis is needed for field measurements. The way we address this is to do the test for several runs repeatedly. Thus the reported correlation is not just a snapshot.

Section 3.4
This is the key portion of the research and should be strongly emphasized. Generally well done.

Line 281: does the statement "....expected values for each aerosol type" directly refer to table 1? If so, reiterate that. If not, please summarize the expected values directly here.

We have added the internal reference (Table 7).

THIS IS ENOUGH TO WORK ON FOR now

---

## Author Response (AR2)

"Review of "Laboratory Validation of a Compact Single-Scattering Albedo (SSA) Monitor" by Perim de Faria et al.

This revised manuscript describes a comparison between the recently developed CAPS PMssa monitor (Aerodyne Research, Inc.) with more established techniques. In response to an earlier review, the authors have clarified that the intent of the manuscript is a comparison, rather than a fundamental evaluation of the performance and accuracy of the instrument. They say, "[This paper does] not address in any explicit way, nor was it designed to address, the question of the absolute uncertainties of the different measurement techniques. It was designed to address the question of how well they correlate." This statement is clearly at odds with the first sentence of the Abstract, which says, "An evaluation of the performance and accuracy of a Cavity Attenuated Phase-Shift Single Scattering Albedo Monitor . . . was conducted. . . .", and, on line 62, "the present optical closure study intends to quantify uncertainties in the measurement of the primary optical properties and the resulting SSA by the CAPS PMssa. . . ". In the "Summary and Outlook" section , the authors state that their analysis has demonstrated an "accuracy of 96% and 99% for the extinction coefficient and scattering coefficient channels." This is not correct; they have instead compared suites of instruments whose accuracies they have neither stated nor evaluated.

Answer: We clarified in the text that we are aiming for relative uncertainties.

Essential to any closure study is a propagation of uncertainties from the raw extinction and scattering measurements through to calculated SSA so that the closure can be quantitatively evaluated. One hopes that, at the end of such a study, a final statement such as, "the measurements agreed with each other within expected uncertainties" can be made. Unfortunately, because errors have not been propagated in the comparisons between the techniques, the usefulness of this multi-instrument comparison is questionable. The instruments certainly appear to agree very well, but is this level of agreement within the "absolute uncertainties" stated in Table 6? Please note that the uncertainties may be lower than the values cited in this table because of the averaging done for this analysis; this needs to be properly accounted for. And are these uncertainties actually propagated from measurement uncertainties, or do they just represent the standard deviations of the measurements? This is not clear.

Answer: We have added the error propagation to the appendix. We have added plots for the CAPS derived properties (SSA and absorption coefficient) as well as for the instrument combination using PSAP and nephelometer showing the relative errors in accuracy for the derived properties (single scattering albedo and absorption coefficient) for the various aerosol types based on 1 second (i.e., unaveraged) data. We did not add the calculated uncertainty data to the scattering plots for visibility reasons. By separating this information in separated plots in Appendix A the dependencies of the relative error on the particles single scattering albedo and the amount of extinction, scattering or absorption is directly visible to the reader. In order to calculate the the uncertainty of the 10 second averaged data a factor of $1/(10)^{.5}=0.316$ has to be applied to the reported 1Hz raw data uncertainties; see Appendix.

In addition to this issue, the authors have not responded to concerns expressed in the first review regarding the slopes of the fitted lines. They pointed out that the lines shown on the graphs were 1:1 lines, not fitted slopes. However, these lines do not pass through the axis origins, so they cannot be 1:1 lines. In addition, the authors did not respond to the concern that the values in the tables did not match regressions performed by the reviewer on the raw data in the supplementary tables. Please confirm that the regressions are correct. The regressions are forced through zero, although there is no justification stated for this choice, and the regressions are one-sided and are not weighted for uncertainties, which are likely heteroscedastic, in the data. This should be rectified.

Answer: The reviewer is right. Revisiting the section it becomes clear that the regression was calculated using 5 minutes averaged values, thus the concern of the reviewer about heteroscedastic is right. We have recalculated all regressions using 10 second averaged data to avoid this issue. In addition we have added Figure 7(new) showing the scatterplot and the associated regression line for one particular case as example.

Line 25 says that the measurement of SSA requires the simultaneous but independent observation of two parameters--extinction and scattering. At line 137, the manuscript says that "the monitor should be thought of as providing separate extinction and SSA values with the scattering channel a derived measurement". This latter statement is incorrect and should be amended.

Answer: done

There are a number of typos and minor wording changes needed in the manuscript; a partial list follows:

**Line 32:** Change "aerosols optical depth" to "aerosol optical depth".
done

**Line 45:** Replace "aerosol optical parameters" with "coefficients". SSA is an "optical parameter" also.
done

**Line 47 and elsewhere:** capitalize only trade names, not instrument types (e.g., integrating nephelometer, cavity ringdown, ammonium sulphate, black carbon should not be capitalized).
done

**Line 70:** The atomizer is a "Collison-type", not "collision-type".
done

**Lines 94-95:** The text describing the roles of the MFCs still does not match Fig. 1.
done

**Line 111 and many places elsewhere in the text:** "data were". "data" is a plural noun.
done

**Lines 123 and 124:** Don't abbreviate "approximately"
done

**Line 125:** Define "LOD".
done (abbreviation deleted (not used further))

**Line 181:** "from the generation system"
done

**Table 3:** I don't understand what the columns "M", "Std m", "B" and "Std b" are. From the caption, there is a "standard deviation of the mean, intercept, standard intercept and R2". What do these mean? What is a "standard intercept"?
An explanation is added to the table description.

**Fig. 5:** The error bars are very small, suggesting that the instruments don't agree within experimental uncertainty. This is because the errors have not been appropriately calculated and propagated. The error bars should be much larger.

**Line 217:** Change "neither" to "either" and "nor" to "or" (because you have "no" in front of "systematic", this is currently a double negative).
Done

**Tables 3 and 4:** Change the column label of "SSA" to "Nominal SSA" to be consistent with the text.

Done

**Fig. 10 caption:** What is "the absorption channel"? There are two lines here. One is from the CAPS PMssa, which does not have an "absorption channel".

Clarified in the text.

**Line 277:** What is meant by, "the deviations being randomly distributed around zero"? Shouldn't they be randomly distributed about the mean of the measured value?

The residuals are distributed around zero.

**Line 284:** Place "relative" in front of "uncertainty".
Done

**Table 6:** The Table caption says "Absolute uncertainty of the SSA measurement". Is this in fact the absolute uncertainty, or is it the standard deviation of the mean value (which is not the same as the uncertainty)?"

Answer: Table 6 compiles the "relative uncertainty of the accuracy" (e.g. Onasch, 2015)

[revised manuscript text omitted]

---

## Author Response (AR3)

**Associate Editor Decision: Publish subject to minor revisions (review by editor)** (12 Aug 2020) by Andrew Sayer
Comments to the Author:
Dear authors,

Thank you for your revised manuscript; I have received one review of it from a referee of the previous version. The referee and I appreciate the additional work put in here, particularly in the estimation of uncertainties. The referee has a number of minor comments on this version (mostly language issues), which I have reproduced below with some notes of my own.

I would be grateful if you could further revise the manuscript to address the below comments. I expect that, after this, I will accept the manuscript for publication in AMT without going out for another round of review. I know that this paper has been in review/revision for somewhat longer than normal and appreciate your patience. These comments are as follows:

I'd like to thank the authors for including a thorough analysis of the uncertainty propagation of the various combinations of instruments, and for making other improvements in the manuscript. I strongly recommend that they incorporate the uncertainties into their discussion and conclusions, focusing on whether agreements between the different instrument combinations agree within these carefully calculated uncertainties. This would be facilitated by plotting these uncertainties, rather than standard deviations, on the x-y plots. With these changes, the manuscript should be acceptable for publication following a substantial number of technical and minor edits. [Editor's note: a few additional sentences, and the plot modifications, would be good here. If the goal is to assess the agreement then in general I think uncertainty is more helpful to show than standard deviation.]

Minor changes and technical edits:

1) Line 23: change "lowest" to "best". "Lowest" is ambiguous; is low accuracy good or bad?

Done , ok

2) Line 29: change "measurement" to "determination". SSA is not directly measured.
Done , ok

3) Line 40: define sigma_ep, sigma_sp, and sigma_ap here; they are not defined in Eq. 1.
Done , ok
4) Lines 45-65, and elsewhere in the manuscript: consistently use the mathematical notation for sigma_ep etc. rather than writing out the variable name each time.
Done , ok

5) Lines 53 and 59: Change "measure" to "determine"

Done , ok

6) Lines 71 and 72: Change "aerosol" to "material"--it's not an aerosol until it's nebulized
Done , ok
7) Lines 113 and 114: Move this sentence to just before Eq. 5.
Done , ok

8) Line 116: Move Eqs. 8 and 9 to just after Eq. 4. You are using Angstrom exponent here; you need to go ahead and define it here.

9) Table 4: The column labels "Blue", "Green" and "Red" make no sense. The row label "Angstrom Exponent" makes no sense. The row label should be "Wavelength Pairs", and the column headings a(B/G) etc. should be replaced with "450 nm/550 nm", "450 nm/700 nm", and "550 nm/700 nm". Get rid of the first row with the unneeded column labels "Blue" etc.

Done , ok

10) Eq. 6: Define sigma_PSAP here, not after Eq. 7.

Already defined there, definition after EQ7 deleted..

11) Eq. 7: Why use omega_naught for SSA when you've been using SSA throughout the text?

These is the original equation.

12) Table 5: Explain what the values in parentheses on the last row are.

done

13) Line 151: Remove "change in". When particles are introduced into the cavity, there is a phase shift, not a change in the phase shift. (I think?)

The change of the phase shift relative to the particle free case (Baseline) is the relevant signal.. Clarified in the text

14) Fig. 2: Make larger. Define LO_out. Use sigma_ep instead of sigma_e, for consistency with the text.

Size will be scaled during production process. Changed to Sigma_ep.

15) Line 172: Negligible differences between what and what?

..clarified in the text

16) Line 181: Change to, "The noise levels were < 1Mm^-1. . . ."

Done

17) Line 182: Replace "lower than" with "<". Also, won't the truncation uncertainty vary with the size of the particles generated? For ambient conditions the uncertainty may be <4%, but maybe not for these lab tests.

Indeed, the truncation error depends on the size distribution. We know the size distribution of the dispersed particles- The truncation corrections is comparable to "ambient" case.

18) Eqs. 8 and 9: There is no need to introduce the variable "w" in Eq. 9; just use "x" and then the two equations make sense with each other (but move both to just after Eq. 4/5).

We think to move these EQs before Eq4/5 would interrupt the "reading" – We have added a link to these Eqs at EQ4/5 instead. (if the definition is needed for the reader)

19) Line 206: Again, this statement is non-sensical. Correlation is not useful and is not a goal; two measurements could be highly correlated yet disagree quantitatively. This study examines if the instrument combinations agree within uncertainties for a range of laboratory aerosols.

OK Agreed I adaped your formulation!

20) Table 6: Undefined parameters SEP, SSP, and SAP are introduced. Please use consistent nomenclature.

Done

21) Line 215: I suggest making the Appendix a Supplemental Materials section instead. [Editor's note: I leave this up to you. Practically, the main difference is that an Appendix will be part of the final journal pdf file while a Supplement will be a separate file also available for download from the final article webpage.]

I think the error propagation section should be part of the paperand ot a separate document. Tus we would leave it like it is.

22) Line 226: "is principle"

Corrected

23) Fig. 4: In the caption explicitly state the time resolution of the data.

Done

24) Line 254: Replace "bellow" with "<".

Done

25) Line 256: Here is a place to clearly state, "These differences are within the combined uncertainties in measurements." Great!

Done!

26) Table 7: Are the values "Std m" and "std b" really standard deviations, or uncertainties in the slope and intercept, respectively?

Reproted is the standared error of the estimate, essentially the expected standard deviation of the calculated value.

27) Line 279: Replace "exclusive" with "purely".
Done!
28) Line 316: Here you refer to Table A3 as a "supplemental table". I agree the appendix should be a "Supplemental Materials".

Changed to "Appendix Table"

29) Line 326: Replace "below" with "<".
Done!
30) Lines 330-337: You may wish to say that the PM_ssa technique measures absorption with uncertainties comparable to that of the PSAP for absorption values greater than some minimum value. Done!

31) Section 3.4: In Fig. 12, the SSA value measured by PM_ssa for the purely scattering aerosol is far outside of the uncertainty value of 3% shown in Table 6. However, the expected uncertainty for a low scattering value is probably worse than 3% (Fig. 13 in the Appendix). In fact, it's hard to reconcile the broad-ranging uncertainties in the SSA values shown in Figs. 13 and 14 with the values in Table 6. Would it not be better to give a range of uncertainties and refer to the figures?

In this figure in particular the standard deviation (variability) is shown by the bars- not the errors of the average itself. The error of the single measurement is in the order +- 8% -9% (Fig. 13) which covers the out layer for low extinction.

32) Generally, I suggest you focus on the level of agreement in Fig. 12 and state if they are within expected uncertainty in the discussion and conclusions. Throughout the manuscript, it would be much better to plot the uncertainty bars (derived from the analysis in the Appendix/Supplemental Materials) in Figs. 5, 9, and 11, rather than the standard deviation of each measurement, to show the level of agreement between the various methods.

I disagree with the latter statement. As long as the instruments show a different time response (e.g. flushtime of the nephelometer > CAPS) it is not only the "numerical error" of the measurement which governs the inter comparison result.  Here also the stability of the experimental conditions (e.g. production) is also needed and represented in the standard deviation to rate the result. Thus, we decided to leave the plots as they are.

33) There are numerous typographical and formatting errors in the references. For example, some references capitalize the title (e.g., Corbin et al.); sometimes the journal name is fully spelled out (e.g. Haywood et al.) and sometimes it's abbreviated (e.g., Heintzenberg et al.). There is an extra carriage return on line 409. These types of errors are characteristic of EndNote-type software; they format irregularly and must always be hand-corrected. [Editor's note: the journal typesetters will give specific queries related to these and apply consistent formatting at the page proofs stage, but it may be worth checking through yourselves as well first.]

Checked..

Please let me know if you have any questions.

Best wishes,

Andrew